ecology, health and disease and epidemiology

zoonotic disease, virus, spillover, threatened species, exploitation, habitat loss

**Author for correspondence:**
Christine K. Johnson
e-mail: ckjohnson@ucdavis.edu

# Global shifts in mammalian population trends reveal key predictors of virus spillover risk

Christine K. Johnson[1], Peta L. Hitchens[2], Pranav S. Pandit[1], Julie Rushmore[1], Tierra Smiley Evans[1], Cristin C. W. Young[1] and Megan M. Doyle[1]

[1]EpiCenter for Disease Dynamics, One Health Institute, School of Veterinary Medicine, University of California, Davis, CA 95616, USA
[2]Melbourne Veterinary School, Faculty of Veterinary and Agricultural Sciences, University of Melbourne, Werribee, VIC 3030, Australia

 CKJ, 0000-0001-6673-8743; PLH, 0000-0002-7528-7056; PSP, 0000-0001-7649-0649; MMD, 0000-0002-3784-355X

Emerging infectious diseases in humans are frequently caused by pathogens originating from animal hosts, and zoonotic disease outbreaks present a major challenge to global health. To investigate drivers of virus spillover, we evaluated the number of viruses mammalian species have shared with humans. We discovered that the number of zoonotic viruses detected in mammalian species scales positively with global species abundance, suggesting that virus transmission risk has been highest from animal species that have increased in abundance and even expanded their range by adapting to human-dominated landscapes. Domesticated species, primates and bats were identified as having more zoonotic viruses than other species. Among threatened wildlife species, those with population reductions owing to exploitation and loss of habitat shared more viruses with humans. Exploitation of wildlife through hunting and trade facilitates close contact between wildlife and humans, and our findings provide further evidence that exploitation, as well as anthropogenic activities that have caused losses in wildlife habitat quality, have increased opportunities for animal–human interactions and facilitated zoonotic disease transmission. Our study provides new evidence for assessing spillover risk from mammalian species and highlights convergent processes whereby the causes of wildlife population declines have facilitated the transmission of animal viruses to humans.

## 1. Introduction

Infectious diseases that originate from animals and infect people comprise the majority of recurrent and emerging infectious disease threats and are widely considered to be one of the greatest challenges facing public health [1–3]. Characterization of pathogen transmission events from wildlife to humans remains an important scientific challenge hampered by pathogen detection limitations in wild species. Disease spillover is probably vastly under-reported, particularly in remote regions where people have limited access to healthcare. Zoonotic disease spillover events are also difficult to detect, especially if the disease spectrum includes mild or non-specific symptoms, or if there is limited to no human-to-human transmission. While the common characteristics of zoonotic diseases have advanced an understanding of disease transmission between animals and humans [4–7], efforts to date have been hampered by sparse data.

The synthesis of epidemiological and ecological profiles of viruses and their hosts has enabled the detection of intrinsic virus and host features linked to species propensity to share viruses with humans [5,8]. For example, host phylogenetic proximity to humans and increased urbanization within a host

**Table 1.** IUCN Red List status and population trend data combined to recategorize species according to conservation status as used for statistical analyses in this study, with number of terrestrial wild mammalian species in each category (*n*).

| Red List status | population trend | conservation status | *n* |
|---|---|---|---|
| critically endangered | combined across all | critically endangered (CR) | 193 |
| endangered | combined across all | endangered (EN) | 439 |
| vulnerable | combined across all | vulnerable (VU) | 493 |
| near threatened | decreasing | near threatened decreasing | 243 |
| near threatened | stable | near threatened stable | 12 |
| near threatened | increasing | near threatened increasing | 7 |
| least concern | decreasing | least concern decreasing | 391 |
| least concern | stable | least concern stable | 1281 |
| least concern | increasing | least concern increasing | 58 |
| data deficient | combined across all | data deficient/unknown trend | 790 |
| least concern | unknown | data deficient/unknown trend | 1371 |
| near threatened | unknown | data deficient/unknown trend | 57 |

distribution has been shown to be positively correlated with the number of zoonotic viruses in a species [5]. Zoonotic disease richness has also been linked to larger geographical range and more litters earlier in life among rodents [9], geographical range overlap and more litters per year among bats [10], and larger body mass, larger geographical range and phylogenetic diversification among carnivores [11].

Characterizing epidemiologic features of viral transmission at the animal–human interface has also revealed a number of high-risk human activities that have enabled virus spillover in the past, particularly in situations that facilitate close contact among diverse wildlife species, domesticated animals and people [4]. Moving from individual circumstances to larger scale drivers requires a historical account of how humans have altered the nature of their contact with animals with implications for zoonotic spillover risk. Domestication of animals, human encroachment into habitats high in wildlife biodiversity and hunting of wild animals have been proposed as key anthropogenic activities driving infectious disease emergence at the global scale [12,13]. Many of these same anthropogenic activities have been implicated as the drivers of wildlife population declines and extinction risk. The International Union for Conservation of Nature (IUCN) Red List of Threatened Species [14] is the authority on global population trends for species, as well as criteria for a species to be listed as threatened with extinction. For the many threatened mammal species, these IUCN metrics provide valuable context for large-scale anthropogenic activities implicated in species declines (e.g. decline in habitat quality for a species), and specific animal–human contact (e.g. exploitation of a species). Here we combine data on all zoonotic viruses detected in terrestrial mammalian species with IUCN metrics on trends in species abundance and threats identified in species declines in order to relate broad-scale patterns in species abundance to spillover risk. By systematically evaluating published data on wild and domesticated mammalian species that have viruses in common with humans, we show that species abundance and specific extinction threats are related to the number of viruses shared with humans across mammalian species, with important implications for understanding virus spillover risk.

## 2. Material and methods

### (a) Zoonotic virus and host datasets

Data were collected from the scientific literature on zoonotic viruses and their terrestrial mammalian hosts published through December 2013. Among 142 zoonotic viruses examined, 139 viruses had at least one mammalian host reported at the species level based on polymerase chain reaction (PCR), virus isolation or serology (electronic supplementary material, Data File S1.) We assumed that detection of a zoonotic virus by PCR or serology indicates the potential for that species to serve as a source of virus spillover to humans, by direct or indirect transmission, in the past or at some point in the future. The number of viruses detected in each mammalian species was summed to estimate zoonotic virus richness for each species. Additional details regarding literature search protocols and data inclusion criteria are provided in the electronic supplementary material.

Data on species abundance, species conservation status and criteria for species listing were obtained from The IUCN 2014 Red List of Threatened Species open source database [14]. The IUCN Red List is the official classifier of species at risk of extinction. This resource includes a list of all mammalian species, Red List categories based on extinction risk, most recently documented population trend (decreasing, stable or increasing), and criteria for listing in IUCN threatened categories, as assessed from 2004 to 2013. There are five categories of Red List status based on extinction risk. For this analysis, two categories of extinction risk, least concern (LC) and near threatened (NT), were expanded into six categories based on IUCN classifications for increasing, decreasing and stable population trend (table 1). Decreasing population trend correlated almost perfectly with population reduction (criterion A) for threatened species, so threatened species were not further categorized according to population trend. Estimates of global abundance were obtained from open sources for humans [15] and domesticated species [16]. Domesticated species were categorized as LC, population increasing.

Criteria used to list species as Threatened by the IUCN Red List [14] provided information on threats faced by wild animal species and reasons for species declines. Several criteria evaluated for wild mammals reflect the potential for human-related impacts, including criterion that indicate likelihood of contact with humans. Criteria and sub-criteria categories that were evaluated statistically for their relationship with zoonotic virus richness observed in each

**Table 2.** Multivariable zero-inflated Poisson regression model predicting the number of zoonotic viruses in mammalian species. (The final zero-inflated Poisson regression model[a] evaluating variation in zoonotic virus richness among extant terrestrial mammalian species is shown with model parameters indicating relative importance (IRR) and significance (with 95% confidence interval) for all variables. Variables significantly associated with the number of zoonotic viruses in a host species included conservation status (as described by the IUCN Red List), criteria for listing of species in a threatened category, taxonomic order, domestication status and (log) number of publications per species in PubMed.)

| variables | IRR[b] | 95% confidence interval | *p*-value |
|---|---|---|---|
| **number of PubMed publications by species (log)** | 1.281 | (1.26, 1.30) | <0.001 |
| **conservation status**[c] | | | |
| least concern increasing | 1.528 | (1.19, 1.95) | 0.001 |
| least concern decreasing | 0.750 | (0.60, 0.94) | 0.011 |
| near threatened decreasing | 0.347 | (0.23, 0.52) | <0.001 |
| vulnerable threatened status | 0.169 | (0.09, 0.30) | <0.001 |
| endangered threatened status | 0.138 | (0.07, 0.25) | <0.001 |
| critically endangered threatened status | 0.076 | (0.03, 0.16) | <0.001 |
| **IUCN criteria for Threatened status**[d] | | | |
| population size reduction by direct observation (A1, A2, A4(a)) | 2.601 | (1.62, 4.21) | <0.001 |
| decline in area of occupancy or habitat quality (A1–A4(c)) | 1.840 | (1.02, 3.31) | 0.042 |
| population size reduction based on levels of exploitation (A1–A4(d)) | 2.28 | (1.36, 3.83) | 0.002 |
| small extent of occurrence (B1) | 0.192 | (0.07, 0.54) | 0.002 |
| **taxonomic order**[e] | | | |
| Primates | 1.363 | (1.13, 1.64) | 0.001 |
| Chiroptera | 2.112 | (1.80, 2.47) | <0.001 |
| Diprotodontia | 0.274 | (0.12, 0.61) | 0.001 |
| Eulipotyphla | 0.192 | (0.10, 0.36) | <0.001 |
| **domesticated species** | 8.051 | (5.89, 11.01) | <0.001 |

[a]Results shown are from the count model (Poisson with log link). The zero-inflation model (binomial with logit link) incorporates the data deficient/unknown population trend variable result as an odds ratio (OR) predicting excess zeros (OR 4.70, 95% CI 3.60–6.13, $p < 0.001$). This zero-inflated Poisson model showed good overall fit (McFadden's $R^2 = 0.247$).

[b]The incident rate ratio (IRR) reflects the relative influence on the expected number of zoonotic viruses in a given species for a given category compared to the reference category specified. This model incorporates a logit model to predict non-detections in host species designated with 'data deficient/unknown population trend'.

[c]Compared to least concern, stable.

[d]Compared to all other criteria for listing as threatened, based on IUCN Red List criteria used to evaluate whether species belong in a threatened category; for threatened species only [14].

[e]Compared to all other orders.

mammalian species are shown in figure 2. Additional details on the criteria and sub-criteria categories assessed are described in the electronic supplementary material.

Analyses were reliant on investigator-driven reports of viruses in animals, which could bias the estimates of zoonotic virus richness in each species, especially if reporting effort was systematically related to risk factors of interest in this study. We incorporated two independent parameters to adjust for potential reporting bias. First, we quantified research publications available in PubMed for each mammalian species in our dataset, and log number of PubMed publications was included in multivariable modelling. Second, we created a data deficient/unknown trend category for each mammalian species in our dataset using IUCN classifications. When there is inadequate information available to make a population assessment, the IUCN classifies some terrestrial mammalian species as data deficient (DD, $n = 790$). In addition, population trend was unknown for many species, including some NT ($n = 57$) and LC species ($n = 1371$). Species lacking enough data to be categorized according to listing criteria, as well as NT and LC species lacking population trend data, were combined into a data deficient/unknown trend category for analyses (table 1). Our assumption for analyses is that species with less population information were potentially less investigated

with respect to zoonotic diseases. Combined measures of threatened status, population trend and data deficiency were summarized as 'conservation status' (table 1).

## (b) Statistical analysis

Correlation between zoonotic virus richness and (i) species richness within taxonomic orders and (ii) abundance estimates for humans and domesticated species were evaluated using Spearmans' $\rho$ statistic for non-parametric variables with a two-tailed test of significance. Multivariable zero-inflated Poisson (ZIP) regression modelling was used to evaluate all putative risk factors for their relationship with zoonotic virus richness (sum of zoonotic viruses) in each mammalian species. Model building was initiated with the log number of PubMed publications, and then variables were entered into the model using forward stepwise entry with all categories of a variable being entered at one time, starting with species status categories, then criteria for listing, then domestication status. Incidence rate ratios (IRRs) and their 95% confidence intervals (CIs) are shown for the final ZIP model (table 2). Stepwise model building procedures are described in more detail in the electronic supplementary material. Parameter importance in improving model fit was assessed by the removal

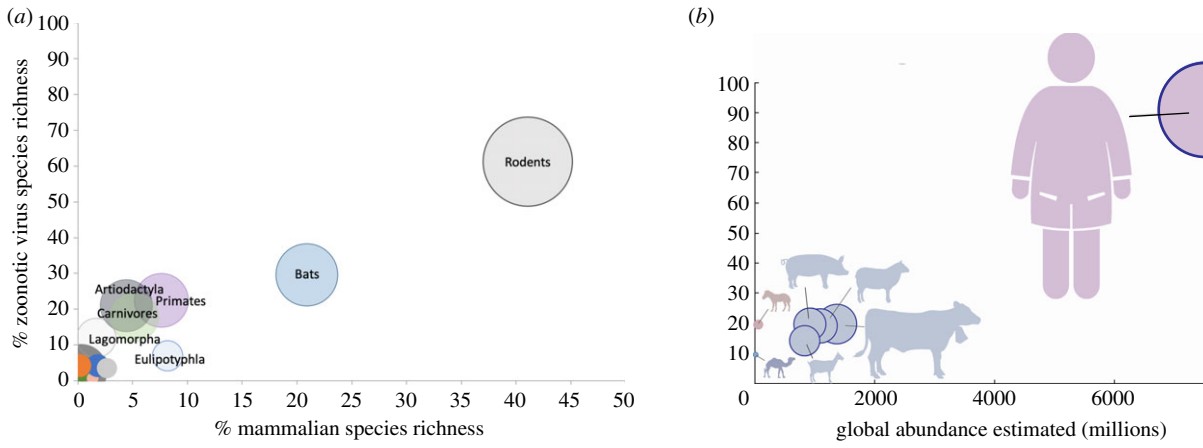

**Figure 1.** Richness of zoonotic viruses found in mammalian hosts, by taxonomic order for wildlife and by species for domesticated animals and humans. (a) Zoonotic virus richness corresponding to species richness among wild mammalian orders. Area of the circles represents the proportion of zoonotic viruses found in species in each order out of the total number of zoonotic viruses among all mammalian species. Orders with less than 5% of zoonotic viruses and less than 2% of mammalian species include Didelphimorphia, Pilosa, Proboscidea, Diprotodontia, Perissodactyla, Cingulata and Dasyuromorphia are not labelled. (b) Zoonotic virus richness corresponding to estimated global abundance (in millions) for humans [15] and domesticated species [16]. Species are coloured according to the order in which they belong in (a). Area of the circles reflects the estimated population size for that species relative to the other species shown. (Online version in colour.)

of parameter groups one at a time, using ΔAkaike information criteria (AIC) (AIC$_{full}$–AIC$_{fitted}$) to compare to the best-fit full model (electronic supplementary material, table S1). We also show the alternate best-fit model, a zero-inflated negative binomial model (electronic supplementary material, table S2), as well as the final ZIP model without the term log number of PubMed publications (electronic supplementary material, table S3) to show model sensitivity to reporting bias.

Statistical analyses were conducted using STATA version 11 SE (StataCorp, College Station, Texas, USA) and the pscl package in R [17,18]. A bipartite (two-mode) affiliation network was generated for virus–host matrix data, stratified by species order. Network data visualization were conducted using the force-directed algorithm FORCEATLAS2 [19] in the software platform GEPHI version 0.9 [20]. All data used to evaluate the relationship between species status, criteria for listing, species order, domestication status, the number of PubMed publications and zoonotic virus richness recognized to the date of the study in a mammalian species are presented in the electronic supplementary material, Data File S2.

## 3. Results and discussion

Global-scale analysis across the breadth of all zoonotic viruses reveals structured variation among mammalian species that have been implicated as a potential source of virus spillover to humans, with predictable patterns in zoonotic virus richness related to species domestication and recent trends in wildlife populations. Among 5335 wild terrestrial mammal species, we found that only 11.4% of mammalian species ($n = 609$) have been identified with one or more of the zoonotic viruses investigated here and, of these, most species (58.1%, $n = 354$) have been reported with only one zoonotic virus each. In line with recent studies [5,21], we found that the highest proportion of zoonotic viruses were reported among species in the orders Rodentia (61%), Chiroptera (30%), Primates (23%), Artiodactyla (21%), Carnivora (18%) and fewer viruses were detected in other mammalian orders (figure 1). Zoonotic virus species richness was highly correlated with mammalian species richness when mammalian host species were grouped by taxonomic order ($\rho = 0.791$, $p < 0.001$), indicating that mammalian orders with more species are the source of more zoonotic viruses (figure 1a), as has been detected in a similar dataset

of zoonotic diseases [21]. We found that three mammalian orders (rodents, bats and primates) have together been implicated as hosts for the majority (75.8%) of zoonotic viruses described to date, and these orders represent 72.7% of all terrestrial mammal species. As a group, domesticated mammals host 50% of the zoonotic virus richness but represent only 12 species. Zoonotic virus richness in domesticated mammalian species was highly correlated with global abundance estimates for humans and domesticated species ($\rho = 0.875$, $p = 0.004$, figure 1b), even when data on humans were dropped from analysis (without humans; $\rho = 0.808$, $p = 0.028$).

The majority (88.6%) of terrestrial mammalian species have not yet been reported with a zoonotic virus, so the ZIP model was fit with 'data deficient' as the variable predicting excess zeros in the data. Holding all other factors in the model constant, an increase in the number of PubMed publications for a species was associated with an increased number of zoonotic viruses reported in that species (table 2). Adjusting for reporting bias prior to the interpretation of other putative factors was important, given publication of zoonotic hosts in the literature was the basis for inclusion in this study, and the inclusion of number of PubMed publications improved model fit as evidenced by change in AIC (electronic supplementary material, table S1). The final ZIP model indicates that conservation status, several criteria for species reductions, taxonomic order and domesticated species status were also significantly related to the number of zoonotic viruses detected in each mammalian species (table 2). Relationships between conservation status, criteria, order, domestication and species richness in zoonotic viruses were robust to alternate model formulations, including zero-inflated negative binomial regression (electronic supplementary material, table S2) and ZIP regression without the term needed to adjust for reporting bias (electronic supplementary material, table S3).

## (a) Zoonotic virus richness scales with wild mammalian abundance

We detected a direct positive relationship between conservation status and the number of viruses shared between that species and humans after adjusting for domestication status, taxonomy,

criteria for listing threatened species, and the number of PubMed publications at the species level (table 2). Less common wildlife species, categorized with increasingly threatened status by the IUCN Red List, were implicated with significantly fewer viruses shared with people, compared to widespread and abundant wild mammalian species. Terrestrial wild animal species of least concern with increasing population trends ($n = 58$) were reported with significantly more zoonotic viruses, while species with decreasing population trends ($n = 391$) had significantly fewer zoonotic viruses, compared to species with stable population trends ($n = 1281$). After adjusting for all factors, we detected a dose-response type relationship between increasingly threatened conservation status and a corresponding decrease in the number of viruses mammals share with humans. The gradual decrease in incidence rate ratios as species abundance declines from least concern conservation status with increasing population trend to critically endangered provides evidence for this trend (table 2). With the exception of species categorized as threatened owing to over-exploitation and habitat loss, this trend can be summarized as follows; species of least concern with increasing abundance were estimated with 1.5 times the number of zoonotic viruses, while species of least concern with decreasing abundance had three-fourths the number of viruses, species not threatened, but decreasing in abundance had one-third the number of viruses, vulnerable species had less than one-sixth the number of viruses, endangered species had one-seventh the number of viruses, and critically endangered had one-thirteenth the number of viruses, compared to species of least concern that were stable in abundance. In an additional analysis of a subset of species that were not found to be data deficient, we found conservation status had a positive linear relationship with the number of zoonotic viruses reported in a species (data shown in the electronic supplementary material).

We found that threatened species listed because of their small extent of occurrence (IUCN Red List category B1, $n = 499$ species) harboured approximately one-fifth as many zoonotic viruses compared to species listed for other reasons when all predictors, including detection bias, were included in the model (table 2). Other IUCN Red List criteria and sub-criteria indicative of small extent of habitat (figure 2) were also correlated with fewer virus detections in a species. In fact, threatened species listed because of very small area of occupancy (IUCN Red List criteria B2), and very small or restricted populations (IUCN Red List criteria D2) have yet to be reported with any zoonotic viruses (figure 2). Previous analyses of parasite richness in primates have found that total parasite richness was lower for species with threatened status, suggesting that small populations with limited geographical range harbour fewer parasites overall [22,23].

Wild mammals with threatened conservation status (i.e. IUCN's Vulnerable, Endangered or Critically Endangered status) are increasingly rare, and the probability of a human encounter is thus presumed to be less likely, unless a species has adapted to human-dominated habitats or is otherwise in frequent contact with humans. Endangered and critically endangered species include many of the most charismatic and intensively managed species in the world, and thus we expected opportunities for virus spillover from species to be more frequent from these species. To further evaluate disparities in zoonotic virus richness among threatened species, we assessed the Red List's listing criteria and sub-criteria in a multivariable modelling approach and found that threatened species for which a population reduction was directly observed (IUCN Red List criteria A1(a), A2 (a) or A4 (a), $n = 53$ species) were predicted to host over 2 times as many zoonotic viruses, compared to species listed as threatened by other means when all other variables were accounted for in the model (table 2). Wildlife populations with declines that have been directly observed were probably more closely monitored to be able to detect changes in population abundance, and often, long-term monitoring programmes accompany species management plans, thereby increasing the likelihood of disease detection and reporting. Also, intensive and often hands-on wildlife management can increase opportunities for pathogen transmission from animals to humans, supporting a biological basis for increased spillover risk beyond increased detection. Direct and indirect contact with wildlife in management and ecotourism settings is a recognized risk for zoonotic spillover, along with increased occupational risk among veterinarians and researchers attending to wildlife [4].

## (b) Convergence in drivers for mammalian species declines and zoonotic virus richness

Among all criteria used to categorize species as threatened with extinction, we identified three additional criteria significantly related to the number of viruses a mammal shares with humans (table 2). After adjusting for other significant effects in the multivariable model, we find that threatened species with a population size reduction owing to exploitation (IUCN Red List category A1–A4(d), $n = 256$ species) have over twice as many zoonotic viruses as compared to threatened species listed for other reasons (table 2). Exploitation of wildlife through hunting and the wild animal trade have been hypothesized as increasing opportunities for pathogen spillover because of the close contact between wildlife and humans involved in these activities [4,12,24,25].

Threatened species with population reductions owing to declines in occupancy, extent of occurrence and/or habitat quality (A1–A4(c), $n = 353$ species) were also predicted to host nearly twice as many zoonotic viruses compared to threatened species declining for other reasons, if all other factors were held constant (table 2). Anthropogenic activities that have altered the landscape, such as forest fragmentation, development and conversion to cropland, have caused declines in wildlife habitat quality, and, as with exploitation, are likely to also increase the probability of animal–human interactions during and subsequent to land conversion activities [26,27]. Human encroachment into biodiverse areas increases the risk of spillover of novel infectious diseases by enabling new contacts between humans and wildlife [28]. Slightly more than half of all threatened species (54.8%) were listed by IUCN because of the impacts of exploitation or habitat loss on species abundance indicating that this is a major impetus for species reductions. Our analysis incorporating data on species declines globally provides broad-scale support for convergent processes whereby exploitation of wildlife and habitat loss have caused wildlife population declines, as well as facilitated the transmission of animal viruses to humans that most likely occurred prior to and during large-scale losses in abundance.

## (c) Domesticated species share the highest number of viruses with humans

Domestication of livestock has played a well-recognized role in transmission of zoonotic viruses to people, as would be

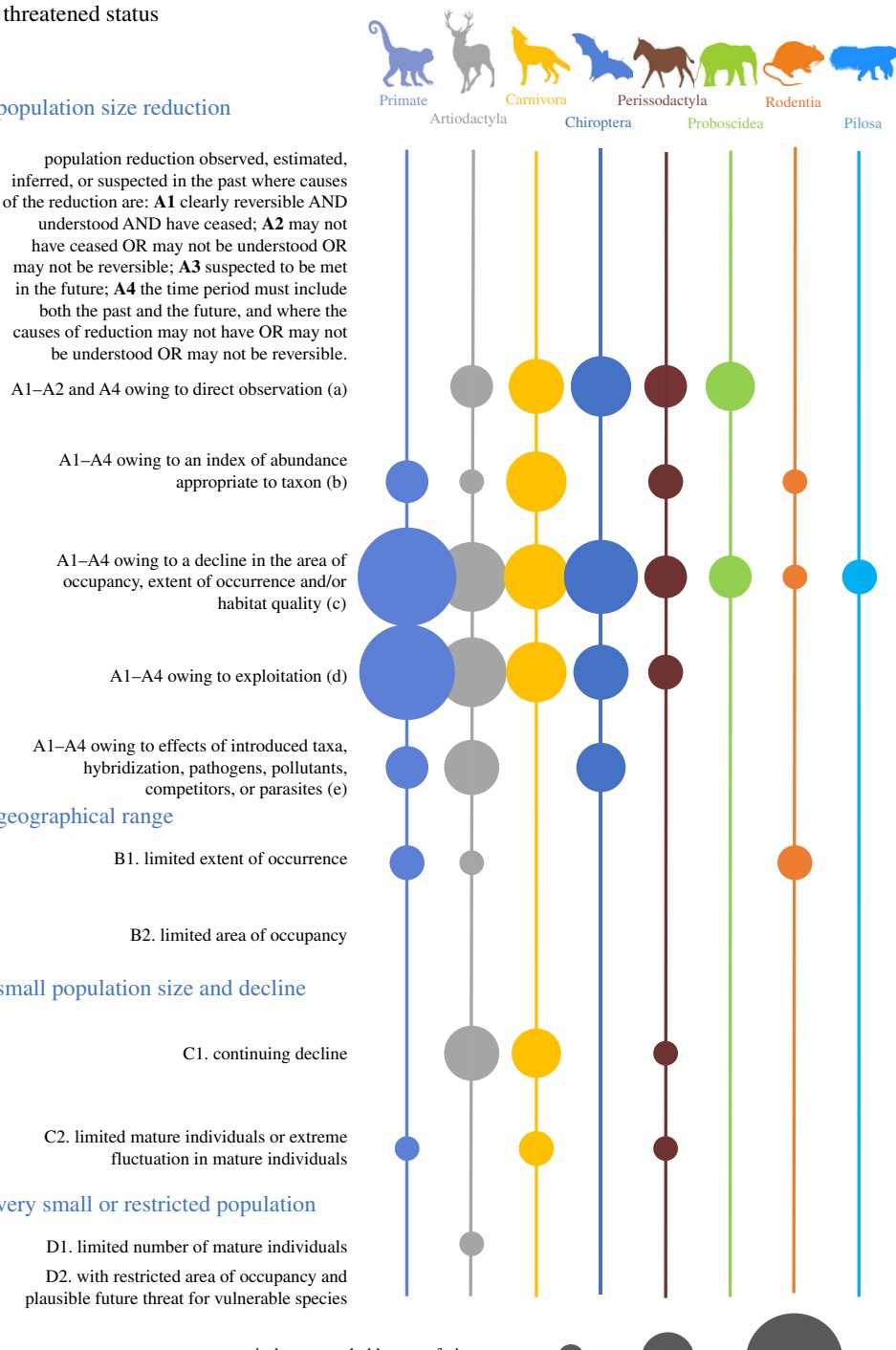

**Figure 2.** Number of mammalian viruses shared with humans for each taxonomic order by IUCN threatened species criteria. The number of zoonotic viruses reported in threatened wildlife species, shown by relative circle area for each taxonomic order according to the scale shown. Scale of circle areas range from one virus (as exemplified by criteria D1 for Artiodactyla) to 16 viruses (as exemplified by criteria A1–A4(c) for primates). Numbers of viruses are not adjusted for factors found to be related to species virus counts in multivariable regression modelling. Species in each order were categorized by the IUCN Red List criteria as adapted for this study. Refer to the IUCN Red List categories and criteria for a detailed explanation of the criteria used by the IUCN to evaluate species trends and place species into threatened categories [14]. (Online version in colour.)

expected of animal species that are unprecedented in their distribution, often reared in dense populations, and have been in close contact with people for centuries [13]. We find that domesticated species status had the largest influence on the number of mammalian viruses shared with humans with eight times more zoonotic viruses predicted in a given domesticated mammal species compared to wild mammalian species

(table 2). Domesticated species harboured an average of 19.3 zoonotic viruses (min 5, max 31) compared to wild species with a mean of 0.23 viruses (min 0, max 16). The top 10 mammalian species with the highest number of viruses shared with humans included eight domesticated species: pigs ($n = 31$ zoonotic viruses), cattle ($n = 31$ zoonotic viruses), horses ($n = 31$ zoonotic viruses), sheep ($n = 30$ zoonotic

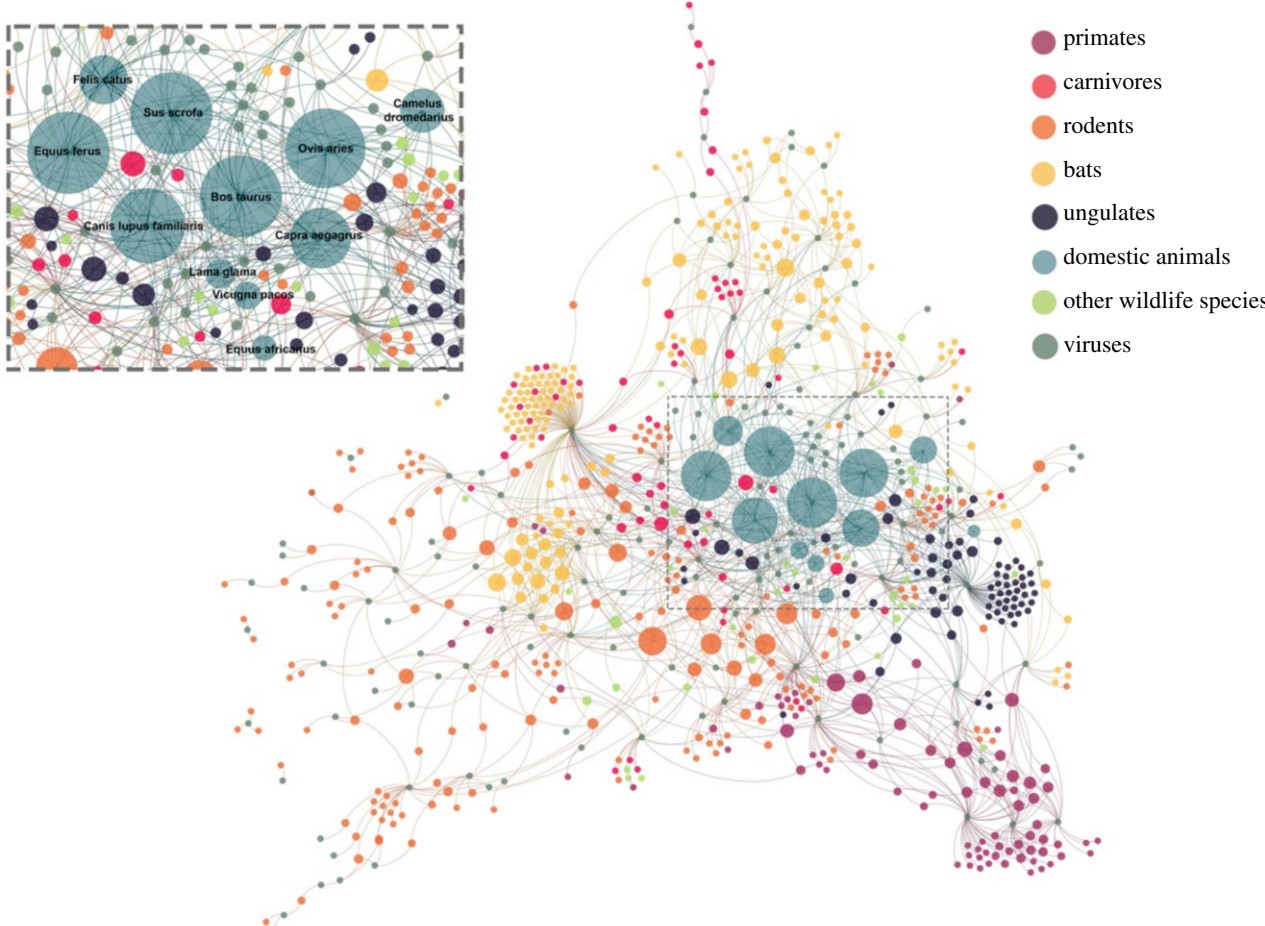

**Figure 3.** Bipartite network showing wild and domesticated mammalian species and their zoonotic virus associations. Host species harbouring the same zoonotic virus are linked by shared zoonotic viruses (grey nodes). Mammalian species nodes are coloured by domestication status and taxonomic order for non-domesticated terrestrial wildlife as shown. Species node size is relative to the zoonotic virus richness calculated in that species. Humans, who are host to all viruses, are not shown. (Online version in colour.)

viruses), dogs ($n = 27$ zoonotic viruses), goats ($n = 22$ zoonotic viruses), cats ($n = 16$ zoonotic viruses) and camels ($n = 15$ zoonotic viruses). Aside from humans, accurate detection and reporting of zoonotic viruses would be most probable in domesticated species, given the economic and public health demand for these data. More accurate detection in domesticated species is supported by the minimal change in estimated number of viruses in regression models with the number of publications (table 2) and without the number of publications as an adjustment for reporting bias (electronic supplementary material, table S3). The only wild animals among the top 10 species with detected zoonotic viruses were the house mouse (*Mus musculus*) and the black rat (*Rattus rattus*), with 16 and 14 zoonotic viruses, respectively. Both of these species in the Rodentia order are considered invasive in most regions of the world, commonly inhabit domestic and peri-domestic structures, and have dubious non-domestication status given their use in laboratory studies and as pets worldwide. Sympatry, or spatial overlap of hosts, was highly correlated with cross-species transmission among rodents, and network analyses illustrate that the global distribution of the house mouse has facilitated the transmission of viruses to sympatric species around the world [29].

Additional support for species domestication as a key feature of increased propensity for sharing viruses with humans is provided by the bipartite network of zoonotic viruses sharing among all mammalian hosts (figure 3). Notably, domestic

animals are among the most central species in the viral sharing network. Viruses in domesticated species were not only commonly shared with other domesticated species but also with wild animal species within respective Cetartiodactyla and Carnivora orders (figure 3). While directionality in historical transmission of viruses between wild mammals and their domesticated kin can only be inferred, we postulate that wild mammals were the original host for the majority of viruses, sharing viruses with domesticated species over centuries of coevolution and domestication. Artiodactyl wild ungulates have been a dominant source of food throughout history and share habitat with domesticated kin. Close phylogenetic relatedness between globally distributed domesticated species and their wild perissodactyl, artiodactyl and carnivore brethren has probably intensified opportunities for cross-species pathogen transmission [30]. Primate, rodent and bat species appear to harbour zoonotic viruses that are not well connected to domesticated species and other wild animal species (figure 3), supporting the premise that these species share zoonotic viruses directly with humans, without domesticated amplifying hosts facilitating viral sharing among species in other orders.

## (d) Primates and bats share more viruses with humans
We found that species in the primate and bat orders were significantly more likely to harbour zoonotic viruses compared to all other orders, after adjusting for domestication, trends in

species abundance, criteria for listing and the number of PubMed publications at the species level (table 2). By contrast, Diprotodontia (marsupials) and Eulipotyphla (shrews, moles, hedgehogs) had fewer zoonotic viruses detected by the time of this study than species in other orders. A recent study evaluating the relationship between phylogeny and the proportion of viruses likely to be zoonotic for a given species also found that bats hosted significantly more zoonotic viruses than other orders and that primates drove the phylogenetic effect as a determinant of zoonotic spillover [5]. The close phylogenetic relationship of humans with non-human primates is recognized as a causal factor underlying spillover, reverse zoonoses and the coevolution of occasionally shared viruses [31]. Bats have also been repeatedly implicated as the source of recent emerging infectious disease events involving high consequence pathogens, including severe acute respiratory syndrome (SARS) [32], Nipah virus encephalitis [33], and hemorrhagic fevers caused by filoviruses [34,35], and have been noted previously to host more zoonotic viruses per species than rodents [10]. Viral sharing has been shown to be more common among bat species than among rodent species and several bat traits have been associated with a higher propensity for cross-species transmission, including gregariousness (roosting in high densities) and migration [29]. With nearly a quarter of bat species lacking sufficient data for categorization of their IUCN Red List status, bats are probably still under-represented in field investigations and warrant future dedicated focus for emerging infectious disease research.

## 5. Conclusion and future directions

Infectious diseases from wildlife have emerged at an increased pace within the last century [36] and are likely to continue to emerge, given expected increases in population growth and landscape change. Curbing disease emergence will prove challenging until we have a more thorough appreciation of the epidemiologic circumstances that facilitate pathogen spillover, particularly from wild animals, which are the source of the majority of recently emerging infectious diseases [2] and continue to constitute a substantial gap in disease detection efforts worldwide. Here, we find broad evidence supporting large-scale mechanisms underlying patterns of zoonotic virus richness across species, by which trends in mammalian abundance and drivers of declines among threatened species reflect animal–human interactions that facilitate virus transmission to people.

By identifying a positive relationship between global trends in mammalian abundance and an increased number of mammalian viruses that have been shared with humans, our findings suggest that mammal species with larger global populations pose greater risk for virus spillover. Our data also provide new evidence that threatened wildlife species with limited extent of occurrence and small population sizes have shared relatively fewer viruses with humans, supporting the concept that virus spillover risk at this large scale is underpinned by the probability of animal–human interactions. Reservoir populations have a critical population or community size required for infectious disease transmission [37], and generally larger populations are more likely to propagate cycles of infection. Population range size similarly reflects opportunities for animal contact, and species with larger ranges should have

increased potential to overlap in range, and possibly share habitat with other species, enabling cross-species transmission and increasing the risk of spillover to humans [29]. However, determinants identified as predictors of zoonotic virus richness at this scale might not relate to zoonotic virus diversity in species at the local scale. Larger population size together with higher population density have been shown to positively correlate with higher viral richness among primate species [22], consistent with disease transmission mechanisms that are dependent on population densities and distributions.

Given we detected a significant increase in zoonotic virus richness among more globally abundant species, additional mechanisms underlying trends in wildlife populations warrant investigation. Species that have increased in abundance and even expanded their range despite large-scale anthropogenically driven landscape change and urbanization [38] are more likely to be generalist species that have adapted to human-dominated landscapes. Approximately one quarter of mammalian species had stable or increasing trends in abundance at the time of analysis, half of which were rodents [14]. While urbanization and landscape change towards crop production could decrease biodiversity overall, these activities can increase the abundance of select species [39]. Many species listed as least concern with increasing abundance by the IUCN Red List are adaptable wild mammalian species that have benefitted from a close relationship with humans. These species could have habitat and dietary niches that overlap with humans in dwellings or in agricultural practices, further enabling direct and indirect contact with similarly adapted sympatric species, domesticated species and humans. In particular, dwellings and agricultural settings are among the most high risk of interfaces for zoonotic viral transmission, particularly from rodents [4]. Pathogen transmission among animals thriving in human-dominated landscapes can also benefit from higher community size and density-dependent viral transmission, especially when resources that sustain mammal populations are aggregated [40], further increasing the probability of human contact with infectious reservoirs in these landscapes. With ongoing landscape transformation towards human-dominated landscapes and approximately half of the world's human population living in urbanized communities [41], species that are adaptable to human modified habitat are likely to continue to be an important source of zoonotic pathogen transmission [40].

Over 20% of mammalian species were threatened with extinction at the time of this analysis, and exploitation and declines in habitat were implicated in the listing status for over half of these threatened species [14]. The IUCN Red List of Threatened Species criteria for categorizing species status [14] was used here to represent large-scale animal–human interactions involved in spillover that could not be measured directly at the species level across all mammalian species. Refined measures of wild animal interactions with people that could constitute effective contact for disease transmission are needed at the local level that can also be scaled up to evaluate broader patterns in spillover risk. We included both serological and molecular data in our analyses, as well as an adjustment for reporting bias, because we were especially concerned about missing host–virus associations. Disease surveillance has been very limited for many wildlife species to date, and wildlife reservoir status can be difficult to ascertain, particularly for viruses with a very short duration of shedding, after which antibodies might only be detectable by serology.

Our model findings were robust to detection bias overall, with the same significant factors explaining variation in species propensity to host zoonotic viruses retaining a similar relative effect and significance even when the number of PubMed publications was not accounted for in the model (electronic supplementary material, table S3). Nonetheless, large-scale surveillance efforts are necessary to more specifically identify epidemiologically relevant animal reservoirs for zoonotic viruses, as well as the periods of heightened shedding that might be related to specific host traits and environmental factors measured at the species level. Wild animal hosts for zoonotic viruses have been vastly under-recognized because the majority of species have not been sampled at the level needed to detect zoonotic viruses, and many geographical regions lack adequate data for modelling [5].

We find evidence to support the premise that abundant mammal species have shared more viruses with humans than less abundant species and that the exploitation of wildlife could have potentiated virus spillover risk. Global patterns in spillover risk reflect close contact interactions between wildlife and humans that occur in a myriad of circumstances around the world. While we shed light on the patterns of zoonotic viruses that have been reported up through the time of this study, we suspect that pathogen spillover often goes unnoticed, with only a proportion of spillover events expanding into outbreaks in people that are subsequently detectable. The evidence of serologic exposure to zoonotic pathogens with high mortality in humans, such as filoviruses, in areas not previously recognized with outbreaks, supports the premise that zoonotic pathogen exposure is more common than recognized [42]. Surveillance for acute febrile illness among people engaged in high-risk activities involving animals, especially wildlife, is a priority to enable more rapid detection of emerging and re-emerging infectious diseases. Surveillance activities that include animals and humans in close contact situations will advance outbreak preparedness in between outbreaks and assist in prioritizing in-depth, longitudinal field studies needed to understand epidemiological patterns in virus transmission and optimize disease prevention actions. Informed mitigation efforts aimed at ensuring biosafety in livestock production, minimizing interactions between wildlife and domesticated animals and limiting close contact with wildlife are especially needed given global trends in urbanization and food production. One Health surveillance approaches are needed that integrate animal and human health in monitoring for emerging infectious diseases and consider environmental change that is likely to intensify close proximity animal–human interactions in the near future.

Data accessibility. The datasets supporting this article have been uploaded as part of the electronic supplementary material.

Authors' contributions. C.K.J. designed the study, analysed data and wrote the manuscript; P.L.H. designed the study, collected data, analysed data and contributed to the manuscript; P.P. analysed data and contributed to the manuscript; J.R. analysed data and contributed to the manuscript; T.S.E. collected data and contributed to the manuscript; C.C.W.Y. analysed data and contributed to the manuscript; M.D. collected data and contributed to the manuscript.

Competing interests. The authors have no competing interests.

Funding. This work was supported by the United States Agency for International Development (USAID) Emerging Pandemic Threat PREDICT program (Cooperative Agreement no. GHN-AOO-09-00010-00). T.S.E. was supported by NIH/Fogarty K01 International Research Scientist Development Award (1K01TW010279-01). The contents of this paper are the responsibility of the authors and do not necessarily reflect the views of USAID or the United States Government.

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
