## [Reviewer comments · Proceedings of the Royal Society B: Biological Sciences]

Review History

RSPB-2019-1561.R0 (Original submission)

Review form: Reviewer 1

Recommendation

Accept with minor revision (please list in comments)

Scientific importance: Is the manuscript an original and important contribution to its field?

Excellent

General interest: Is the paper of sufficient general interest?

Excellent

Quality of the paper: Is the overall quality of the paper suitable?

Good

Is the length of the paper justified?

Yes

Should the paper be seen by a specialist statistical reviewer?

Yes

Do you have any concerns about statistical analyses in this paper? If so, please specify them explicitly in your report.

No

It is a condition of publication that authors make their supporting data, code and materials available - either as supplementary material or hosted in an external repository. Please rate, if applicable, the supporting data on the following criteria.

Is it accessible?

Yes

Is it clear?

Yes

Is it adequate?

Yes

Do you have any ethical concerns with this paper?

No

Comments to the Author

Overall, I enjoyed reading this work, titled “Global shifts mammalian declines reveal key predictors of virus spillover risk” by Johnson and colleagues. Personally, I feel this work addresses an important problem, the authors introduce the problem well, use established methods, present their findings clearly, and discuss the weaknesses of the work fairly. I was a little skeptical as to how they would use the IUCN data and how ‘novel’ their framework was [see abstract], as these data have significant weaknesses. However, I thought the authors used them well and quite imaginatively to come to conclusions that had until recently largely been discussed, but not rigorously analysed in the literature.

Major issues

Data:

The methods are fair and well explained and overall I agree with the decisions made and/or the authors have given sound justification for their decisions.

However, there is one potential exception, and that is that several viruses I see cited are not actually known to be zoonotic, but are potentially zoonotic. For example, the bat lyssaviruses Lagos bat virus, Shimoni bat virus, West Caucasian bat virus and Aravan virus have not, as far as I am aware, been isolated from people. Most of the others have, but for accuracy these either need excluding or the authors need to justify why they are included. My guess is that excluding these won’t change the results, but you should check.

Presentation:

After reading all the supplementary information, I believe that the methods are appropriate, but it was hard to follow. It wasn’t really until the Statistical analysis section in the SI methods (starting on line 68) that I really understood how the authors had reached the modeling approach that they presented. In the details below I make some suggestions, but overall, I would prefer more of the statistical analyses text to be in the main text, particularly relating to the ZIP regression model selection and choice. This seems essential for the reader.

Details

Methods

IUCN data - lines 99 to 119: I know this section is key to this, but I found this quite hard to follow. I wonder if some of this can be moved to the SI and a simpler version (perhaps an adaptation of SI Table S2) included in the main text?

Statistical analyses - I preferred the presentation of the methods in the SI. Key factors that lead the authors to using this regression model are not included here and I think that they are important. Also, it was really clear to me until the results, line 159 why a zero inflated model was even being used on my first read, and that non-zoonotic virus hosts were included.

Results

Line 185-9 - this seems a crucial result, perhaps present the model structure in the main text?

Line 253: "...activities." In the other sections these comments are always followed by references, but not here. I would recommend the following:

Hahn, et al (2014) "Roosting behaviour and habitat selection of *Pteropus giganteus* reveal potential links to Nipah virus epidemiology" *J Appl Ecol*, 51: 376-387

Hahn, et al (2014) "The Role of Landscape Composition and Configuration on *Pteropus giganteus* Roosting Ecology and Nipah Virus Spillover Risk in Bangladesh." *Am J Trop Med Hyg*, 90(2): 247 - 255.

Rulli, et al (2017) "The nexus between forest fragmentation in Africa and Ebola virus disease outbreaks." *Scientific reports* 7 (2017): 41613.

Lines 327 & 377: In both instances, but especially 377 I think the work by Wilkinson et al is relevant, because they develop an explicit model of habitat fragmentation and infectious disease emergence risk:

Wilkinson, et al (2018) "Habitat fragmentation, biodiversity loss and the risk of novel infectious disease emergence" *J Roy Soc Interface* 15 (149), 20180403

Lines 334 - 346: here, critical community size is alluded to, but never really mentioned. The focus appears to be on population density, but some (maybe most?) wildlife species don't necessarily increase their density in the same spatial location with increased abundances, as my guess is that many species have behavioural traits (including nutrition availability and competition) that limit species densities. Maybe refer explicitly to critical community sizes, whilst also acknowledging that other transmission processes exist (e.g. frequency dependent transmission), and see:

Lloyd-Smith, J.O. et al. (2005) "Should we expect population thresholds for wildlife disease?" *Trends in Ecology & Evolution* 20: 511-519

SI Line 31-32: serology is more sensitive, but not in 'virus detection', as typically for acute infectious diseases this is historic infection. I recognise this is not true for some persistent infections and used for rodent arenaviruses (for example), but please correct this.

Review form: Reviewer 2

Recommendation

Accept with minor revision (please list in comments)

Scientific importance: Is the manuscript an original and important contribution to its field?

Excellent

General interest: Is the paper of sufficient general interest?

Excellent

Quality of the paper: Is the overall quality of the paper suitable?

Good

Is the length of the paper justified?

Yes

Should the paper be seen by a specialist statistical reviewer?

Yes

Do you have any concerns about statistical analyses in this paper? If so, please specify them explicitly in your report.

No

It is a condition of publication that authors make their supporting data, code and materials available - either as supplementary material or hosted in an external repository. Please rate, if applicable, the supporting data on the following criteria.

Is it accessible?

Yes

Is it clear?

Yes

Is it adequate?

Yes

Do you have any ethical concerns with this paper?

No

Comments to the Author

Johnson et al present an original statistical analysis of conservation status (IUCN database) and carriage of 143 zoonotic viruses in 5,438 wild mammalian species. Overall, more endangered species tend to have fewer zoonotic viruses, and wild mammals have fewer zoonotic viruses than domesticated ones. Among threatened species, those that suffer from habitat loss or direct exploitation have more zoonotic viruses than others.

These results are interesting and important as they cast light on the complex relationship between two facets of the human-wildlife relationship: conservation and zoonotic spillover. These two issues are of major concern globally and have been in direct conflict in several occasions in recent years, when zoonotic spillover from wildlife has triggered culling campaigns opposed by conservationists. As illustrated by the controversies around flying foxes and Hendra virus in Australia, badgers and bovine TB in Great Britain, or bats and rabies in South America, to cite but a few, cultural and sociological factors exacerbate these conflicts and prevent their resolution, with both sides cherry-picking the scientific evidence and economic factors that align with their

beliefs. What this study brings is a dispassionate and global analysis of the relationship that exist between the occurrence of zoonotic viruses and conservation status. The inclusion of sources of threat enables the authors to propose a mechanistic interpretation of the statistical association. Even though precise causation and reliable solutions are beyond the scope of this analysis, the results presented here will be very valuable to inform the debate and guide more detailed investigations of the processes underlying zoonotic spillover and wildlife population variations.

I have no major issues with this manuscript. Although I'm not qualified to properly assess the statistical analyses carried out by the authors, I know from previous experience that this type of analysis is always imperfect and comes with caveats, especially in relation to the various flaws and biases in the databases. I think the authors are quite open about these issues and have been cautious in not over-interpreting their findings.

Specific comments:

- L. 40-42: I found this sentence a bit awkward ("viruses have been linked to pandemic properties") and I'm not sure it's needed.
- L.70-73: the proxy argument seems a bit tenuous here without further explanation. You don't need to offer this proxy to justify or motivate the analysis. I would suggest rephrasing this sentence.
- L.87: why only until December 2013? Given the recent increase in zoonotic virus research (not least from the PREDICT initiative), I suspect that the data may have increased substantially in the last 5 years.
- L.133: is that a fair assumption, given that zoonotic viruses are often reported in very small numbers of wild animals within species? Could this be tested in some way? Given the authors' involvement in one of the largest wildlife screening initiatives to date (PREDICT), I expect they would have access to some form of evidence for or against this claim.
- L. 266: is that a number of viral species? More generally, what taxonomic unit was used to count zoonotic viruses?

Review form: Reviewer 3

Recommendation

Major revision is needed (please make suggestions in comments)

Scientific importance: Is the manuscript an original and important contribution to its field?

Good

General interest: Is the paper of sufficient general interest?

Good

Quality of the paper: Is the overall quality of the paper suitable?

Marginal

Is the length of the paper justified?

Yes

Should the paper be seen by a specialist statistical reviewer?

Yes

Do you have any concerns about statistical analyses in this paper? If so, please specify them explicitly in your report.

Yes

It is a condition of publication that authors make their supporting data, code and materials available - either as supplementary material or hosted in an external repository. Please rate, if applicable, the supporting data on the following criteria.

Is it accessible?

Yes

Is it clear?

Yes

Is it adequate?

No

Do you have any ethical concerns with this paper?

No

Comments to the Author

Johnson et al undertakes to assess links between mammalian abundance and IUCN threat category with the detection of zoonotic viruses.

This work includes a significant and valuable effort to create a dataset on the known viruses detected in mammals. I am aware that this has been done previously for rodents, bats and smaller groups of taxa, but if this is really the first database of its kind for mammals, then it should be clarified further within the manuscript and celebrated as a contribution in its own right. If it builds off or expands previous similar datasets (perhaps Han et al, as is mentioned in the results), then this should also be clarified.

The analyses rest on the assumption that IUCN species population metrics and threat measures are a proxy for the likelihood of animal- human interactions. The manuscript feels like the analysis have been run without any particular hypotheses of how these interactions might be occurring under different circumstances e.g. common but rapidly declining species vs rarer but stable species. Other presumably important factors are not considered, for example, how habitat type and habitat/dietary specialisation might correlate with human contact. Similarly, urbanisation favours survival of generalist over specialist species, and this is likely reflected in human contacts (more contacts for generalist species that thrive in urban environments), even if their overall population metrics and threat measures remain the same. The focus of the text seems to switch between which groups of species "harbour the most zoonotic viruses" versus which are of greater spillover risk, and I argue that these are not the same things. Detectability and contact rates with humans will both play a significant role.

I feel that the analyses are of considerable interest, however I am not currently convinced by the implementation and interpretation of the statistical analyses. In describing the analyses, the relative importance of each of the categories is unclear and the final model is not reported. For example, with most of the explanatory power consumed by research effort and domestication status, how much real contribution to zoonotic virus risk are the IUCN categories and trends actually contributing? Much of the discussion focusses on this rather than the model terms with higher importance in the model.

I have made more specific comments below

Specific comments:

Line 18 - We don't know how many zoonotic viruses - are in mammalian species. Change to 'detected in'. Similarly for line 20 - 'have more viruses detected'

Line 21 - A sentence is required inserted in line 21 that describes the proposed mechanisms. Also, there is no mention of domestic animals or research effort in the abstract.

Line 28 - Clarify whether the first half of this sentence is referring only to humans. I don't think that zoonotic diseases comprise the majority of emerging infectious disease threats to wildlife (consider Bd in amphibians, WNS in North American bats, DFTD in Tasmanian devils, Avian Malaria, Canine Distemper etc).

Lines 28- 55: This is a long paragraph with lots of ideas that don't easily flow. I feel that it could be edited to be more concise and more quickly get to the point. e.g. lines 32-38.

Also, to attract the broader readership that is expected for Proc B, I suggest including a few words to define zoonotic.

Lines 38-42 - is the focus of the paper on spillover or pandemic potential? There is no further mention of pandemics through the text and the focus on pandemic potential seems irrelevant. I suggest either rephrasing in terms of spillover potential or deleting.

Lines 49 - 51 - clarify whether high or low body mass or human population density increases/decreases propensity to share viruses with humans

Line 51 - high risk as determined by what metric? This is a bit confusing. Suggest deleting "Within specific high risk taxa" from sentence

Line 52 - Replace 'numbers' with 'detections' or similar. We have no idea how many viruses most species have - can only judge what has been detected.

Lines 71-73 - As discussed in general comments above, this assumption requires further justification from an ecological perspective.

Lines 73- 74 Provide an example here of the types of contact being referred to

Line 75 - this is the first time that the focus has shifted to mammals. Provide justification for this.

Also clarify here that the focus is on terrestrial species

Line 80-81- the results of the study don't have implications for the the risk of spillover per se - it has implications for understanding the risk, or for managing the risk.

Line 93-102 - A table would be helpful to summarise the IUCN threat categories and how they link with assumed abundance. It is unclear what "criteria A1-4 by sub-criteria a, b, c, d, e" means, for example. I got completely lost in the remainder of this paragraph

Line 153 - In general, the authors should be cautious of saying Species with X characteristic have more zoonotic viruses. Despite trying to account for biases, there are still detection issues there, and phrasing should reflect that (as suggested in the abstract)

Line 159 - should this be "one or more of the zoonotic viruses"

Line 160 - (58.8%, n= 84)

Line 161 - quantify what is meant by 'vast majority'

Figure 1 - It is unclear which circles the lines are pointing to in B.

Line 164 - 167 - Clarify whether both viruses and hosts or just hosts were grouped by taxonomic order.

Lines 167-169 - Clarify what proportion of all terrestrial mammalian species these orders represent.

Table 1: In assessing the effect of "Least Concern increasing", it is unclear in the text whether includes or accounts for domestic species or not (lines 195-197).

Table 1: Suggest present this from top to bottom in order of importance -i.e. as outlined in lines 187-189. Also, the final model structure is not given, and it is unclear whether the effects presented take into account the hierarchical structure of the data e.g. only threatened species are classified under "IUCN Criteria for Threatened Status"

Line 303 - 304 - The wording here is helpful to clarify that effects being reported take into account other effects, but this is not mentioned elsewhere.

Figure 2: It took me a while to see that the lines were connecting the animal silhouettes to the relevant circles for that species. A minor edit that might better highlight this would be to delete the grey lines after they pass (i.e. above) the silhouettes so it looks more like a line connecting to the silhouette.

It is also confusing to have these silhouettes overlapping with the (long) Population size reduction text. I suggest abbreviating this text to a few words in the figure, and have the full description in the Figure caption. I think it would also assist digestibility of the figure if the four categories in blue were also labelled A-D.

Lines 276 - 321 - Much of the text in this section is discussion, not results. It should be moved to the discussion.

Discussion - By this point, taking into account the statement on lines 187 - 189 and the content in table 1, the take-home message I have inferred is that the research effort is the main determinant for detection of zoonotic viruses, followed by whether the species is domestic or not.

Domesticated species contribute far more zoonoses than terrestrial mammalian wildlife species (lines 262-267). This should be the main finding, and any mention of the smaller effects of population size or IUCN status (which the authors are putting forward as the main finding) should be placed within that context. Of the fraction of zoonotic viruses that come from wildlife, significant effects of population size and threat status are evident, but how many viruses/how much risk does this represent in real terms?

Lines 334-340 - Species distribution, human distribution and habitats are not taken into account. The authors need to demonstrate the link between large population sizes and human contact, or explicitly state that this is a foundational assumption of these analyses. This is partly addressed in lines 341 - 342, but range overlap does not equate with habitat overlap

Line 405-406: I do not agree that there is evidence to support that "abundant mammal species harbor more zoonotic viruses than less abundant species". There has been greater opportunity for those to spillover into people, however the results for the threat categories show that that certain circumstances just allow us to be aware of the zoonotic viruses present in less abundant species.

Line 409 - 410: Given the majority of zoonotic viruses come from domestic animals to humans, how much effect will this provide versus simply focussing on domestic animal health? There is no mention of the force of infection from domestic animals here.

Additionally, I accept that it is difficult to study, and outside of the scope of this manuscript, but the number of zoonotic viruses from any given species does not reflect the burden of disease in people. The need for this information should be highlighted in the discussion.

Supplementary methods

Lines 8-10 - For completeness, add a date as to when the search was conducted, and the range of years of studies returned. Also, in the search terms, were wildcards included? e.g. zoonos* (to catch zoonosis and zoonoses). Was this list cross checked against existing databases (e.g. those on bat and rodent viruses) to see whether any were missed with this approach?

Lines S28-30 - Although it is stated in the supplementary materials, clarify here that evidence included PCR, virus isolation and serological evidence

Lines S69 - It is helpful if the Supplementary methods are readable on their own - Explain what the 'ZIP model' is.

Decision letter (RSPB-2019-1561.R0)

15-Aug-2019

Dear Dr Johnson:

I am writing to inform you that your manuscript RSPB-2019-1561 entitled "Global shifts in mammalian declines reveal key predictors of virus spillover risk" has, in its current form, been rejected for publication in Proceedings B.

This action has been taken on the advice of referees, who have recommended that substantial revisions are necessary. With this in mind we would be happy to consider a resubmission, provided the comments of the referees are fully addressed. However please note that this is not a provisional acceptance.

The resubmission will be treated as a new manuscript. However, we will approach the same reviewers if they are available and it is deemed appropriate to do so by the Editor. Please note

that resubmissions must be submitted within six months of the date of this email. In exceptional circumstances, extensions may be possible if agreed with the Editorial Office. Manuscripts submitted after this date will be automatically rejected.

Sincerely,
 Professor Hans Heesterbeek
 mailto: proceedingsb@royalsociety.org

Associate Editor

Board Member: 1

Comments to Author:

It's clear all referees found the manuscript interesting, and that it contributes valuable results to an important field- a view which I concur with also. There are many minor points raised by the referees, but focussing on the more key points:

Referee 1 raises the issue that perhaps some viruses included are only potentially zoonotic and this should be checked.

Referee 1 also raises that main text statistical methods are not easy to follow, but that the SI is necessary for the reader to get a grip on what was actually done.

Ref 3 raises a number of points, but they pertain mainly to the presentation of the results, and deductions that follow from these. There are a lot of specific points drawn out for clarification, but see the comments on figure 2 in particular.

A few minor points on the figures from me:

- Fig 1: what is the area of the circles representing?
- Fig 2: when it says "size" of circles for this figure, is it area? Also give some scale for these.
- Fig 2: I agree with ref 3 that it took a while to see what's going on here. In addition to ref 3's suggestions, eliminate the extra grey line at the left, reduce the text through the figure. Consider making the grey lines match to the species colours. Consider putting the species icons on the same row (make the text vertical).
- Fig 2: what is going on with perissodactyla?? They look a bit outlier and this isn't referred to in main text.

Bringing together the views of the referees, it's clear the main text could be tightened up in places, and some of the key methods from SI could be brought in to make more clear what was actually done. Referee 2 does comment that the authors have been cautious in not over-interpreting findings, but perhaps the interpretation could be reduced or streamlined (there is some repetition).

Reviewer(s)' Comments to Author:

Referee: 1

Comments to the Author(s)

Overall, I enjoyed reading this work, titled "Global shifts mammalian declines reveal key predictors of virus spillover risk" by Johnson and colleagues. Personally, I feel this work addresses an important problem, the authors introduce the problem well, use established methods, present their findings clearly, and discuss the weaknesses of the work fairly. I was a little skeptical as to how they would use the IUCN data and how 'novel' their framework was [see abstract], as these data have significant weaknesses. However, I thought the authors used them well and quite imaginatively to come to conclusions that had until recently largely been discussed, but not rigorously analysed in the literature.

Major issues

Data:

The methods are fair and well explained and overall I agree with the decisions made and/or the authors have given sound justification for their decisions.

However, there is one potential exception, and that is that several viruses I see cited are not actually known to be zoonotic, but are potentially zoonotic. For example, the bat lyssaviruses Lagos bat virus, Shimoni bat virus, West Caucasian bat virus and Aravan virus have not, as far as I am aware, been isolated from people. Most of the others have, but for accuracy these either need excluding or the authors need to justify why they are included. My guess is that excluding these won't change the results, but you should check.

Presentation:

After reading all the supplementary information, I believe that the methods are appropriate, but it was hard to follow. It wasn't really until the Statistical analysis section in the SI methods (starting on line 68) that I really understood how the authors had reached the modeling approach that they presented. In the details below I make some suggestions, but overall, I would prefer more of the statistical analyses text to be in the main text, particularly relating to the ZIP regression model selection and choice. This seems essential for the reader.

Details

Methods

IUCN data - lines 99 to 119: I know this section is key to this, but I found this quite hard to follow. I wonder if some of this can be moved to the SI and a simpler version (perhaps an adaptation of SI Table S2) included in the main text?

Statistical analyses - I preferred the presentation of the methods in the SI. Key factors that lead the authors to using this regression model are not included here and I think that they are important. Also, it was really clear to me until the results, line 159 why a zero inflated model was even being used on my first read, and that non-zoonotic virus hosts were included.

Results

Line 185-9 - this seems a crucial result, perhaps present the model structure in the main text?

Line 253: "...activities." In the other sections these comments are always followed by references, but not here. I would recommend the following:

Hahn, et al (2014) "Roosting behaviour and habitat selection of *Pteropus giganteus* reveal potential links to Nipah virus epidemiology" *J Appl Ecol*, 51: 376-387

Hahn, et al (2014) "The Role of Landscape Composition and Configuration on *Pteropus giganteus* Roosting Ecology and Nipah Virus Spillover Risk in Bangladesh." *Am J Trop Med Hyg*, 90(2): 247 - 255.

Rulli, et al (2017) "The nexus between forest fragmentation in Africa and Ebola virus disease outbreaks." *Scientific reports* 7 (2017): 41613.

Lines 327 & 377: In both instances, but especially 377 I think the work by Wilkinson et al is relevant, because they develop an explicit model of habitat fragmentation and infectious disease emergence risk:

Wilkinson, et al (2018) "Habitat fragmentation, biodiversity loss and the risk of novel infectious disease emergence" *J Roy Soc Interface* 15 (149), 20180403

Lines 334 - 346: here, critical community size is alluded to, but never really mentioned. The focus appears to be on population density, but some (maybe most?) wildlife species don't necessarily increase their density in the same spatial location with increased abundances, as my guess is that many species have behavioural traits (including nutrition availability and competition) that limit species densities. Maybe refer explicitly to critical community sizes, whilst also acknowledging that other transmission processes exist (e.g. frequency dependent transmission), and see:

Lloyd-Smith, J.O. et al. (2005) "Should we expect population thresholds for wildlife disease?" *Trends in Ecology & Evolution* 20: 511-519

SI Line 31-32: serology is more sensitive, but not in 'virus detection', as typically for acute infectious diseases this is historic infection. I recognise this is not true for some persistent infections and used for rodent arenaviruses (for example), but please correct this.

Referee: 2

Comments to the Author(s)

Johnson et al present an original statistical analysis of conservation status (IUCN database) and carriage of 143 zoonotic viruses in 5,438 wild mammalian species. Overall, more endangered species tend to have fewer zoonotic viruses, and wild mammals have fewer zoonotic viruses than domesticated ones. Among threatened species, those that suffer from habitat loss or direct exploitation have more zoonotic viruses than others.

These results are interesting and important as they cast light on the complex relationship between two facets of the human-wildlife relationship: conservation and zoonotic spillover. These two issues are of major concern globally and have been in direct conflict in several occasions in recent years, when zoonotic spillover from wildlife has triggered culling campaigns opposed by conservationists. As illustrated by the controversies around flying foxes and Hendra virus in Australia, badgers and bovine TB in Great Britain, or bats and rabies in South America, to cite but a few, cultural and sociological factors exacerbate these conflicts and prevent their resolution, with both sides cherry-picking the scientific evidence and economic factors that align with their beliefs. What this study brings is a dispassionate and global analysis of the relationship that exist between the occurrence of zoonotic viruses and conservation status. The inclusion of sources of threat enables the authors to propose a mechanistic interpretation of the statistical association.

Even though precise causation and reliable solutions are beyond the scope of this analysis, the results presented here will be very valuable to inform the debate and guide more detailed investigations of the processes underlying zoonotic spillover and wildlife population variations.

I have no major issues with this manuscript. Although I'm not qualified to properly assess the statistical analyses carried out by the authors, I know from previous experience that this type of analysis is always imperfect and comes with caveats, especially in relation to the various flaws and biases in the databases. I think the authors are quite open about these issues and have been cautious in not over-interpreting their findings.

Specific comments:

- L. 40-42: I found this sentence a bit awkward ("viruses have been linked to pandemic properties") and I'm not sure it's needed.
- L.70-73: the proxy argument seems a bit tenuous here without further explanation. You don't need to offer this proxy to justify or motivate the analysis. I would suggest rephrasing this sentence.
- L.87: why only until December 2013? Given the recent increase in zoonotic virus research (not least from the PREDICT initiative), I suspect that the data may have increased substantially in the last 5 years.
- L.133: is that a fair assumption, given that zoonotic viruses are often reported in very small numbers of wild animals within species? Could this be tested in some way? Given the authors' involvement in one of the largest wildlife screening initiatives to date (PREDICT), I expect they would have access to some form of evidence for or against this claim.
- L. 266: is that a number of viral species? More generally, what taxonomic unit was used to count zoonotic viruses?

Referee: 3

Comments to the Author(s)

Johnson et al undertakes to assess links between mammalian abundance and IUCN threat category with the detection of zoonotic viruses.

This work includes a significant and valuable effort to create a dataset on the known viruses detected in mammals. I am aware that this has been done previously for rodents, bats and smaller groups of taxa, but if this is really the first database of its kind for mammals, then it should be clarified further within the manuscript and celebrated as a contribution in its own right. If it builds off or expands previous similar datasets (perhaps Han et al, as is mentioned in the results), then this should also be clarified.

The analyses rest on the assumption that IUCN species population metrics and threat measures are a proxy for the likelihood of animal- human interactions. The manuscript feels like the analysis have been run without any particular hypotheses of how these interactions might be occurring under different circumstances e.g. common but rapidly declining species vs rarer but stable species. Other presumably important factors are not considered, for example, how habitat type and habitat/dietary specialisation might correlate with human contact. Similarly, urbanisation favours survival of generalist over specialist species, and this is likely reflected in human contacts (more contacts for generalist species that thrive in urban environments), even if their overall population metrics and threat measures remain the same. The focus of the text seems to switch between which groups of species "harbour the most zoonotic viruses" versus which are of greater spillover risk, and I argue that these are not the same things. Detectability and contact rates with humans will both play a significant role.

I feel that the analyses are of considerable interest, however I am not currently convinced by the implementation and interpretation of the statistical analyses. In describing the analyses, the

relative importance of each of the categories is unclear and the final model is not reported. For example, with most of the explanatory power consumed by research effort and domestication status, how much real contribution to zoonotic virus risk are the IUCN categories and trends actually contributing? Much of the discussion focusses on this rather than the model terms with higher importance in the model.

I have made more specific comments below

Specific comments:

Line 18 - We don't know how many zoonotic viruses are in mammalian species. Change to 'detected in'. Similarly for line 20 - 'have more viruses detected'

Line 21 - A sentence is required inserted in line 21 that describes the proposed mechanisms. Also, there is no mention of domestic animals or research effort in the abstract.

Line 28 - Clarify whether the first half of this sentence is referring only to humans. I don't think that zoonotic diseases comprise the majority of emerging infectious disease threats to wildlife (consider Bd in amphibians, WNS in North American bats, DFTD in Tasmanian devils, Avian Malaria, Canine Distemper etc).

Lines 28- 55: This is a long paragraph with lots of ideas that don't easily flow. I feel that it could be edited to be more concise and more quickly get to the point. e.g. lines 32-38.

Also, to attract the broader readership that is expected for Proc B, I suggest including a few words to define zoonotic.

Lines 38-42 - is the focus of the paper on spillover or pandemic potential? There is no further mention of pandemics through the text and the focus on pandemic potential seems irrelevant. I suggest either rephrasing in terms of spillover potential or deleting.

Lines 49 - 51 - clarify whether high or low body mass or human population density increases/decreases propensity to share viruses with humans

Line 51 - high risk as determined by what metric? This is a bit confusing. Suggest deleting "Within specific high risk taxa" from sentence

Line 52 - Replace 'numbers' with 'detections' or similar. We have no idea how many viruses most species have - can only judge what has been detected.

Lines 71-73 - As discussed in general comments above, this assumption requires further justification from an ecological perspective.

Lines 73- 74 Provide an example here of the types of contact being referred to

Line 75 - this is the first time that the focus has shifted to mammals. Provide justification for this. Also clarify here that the focus is on terrestrial species

Line 80-81- the results of the study don't have implications for the the risk of spillover per se - it has implications for understanding the risk, or for managing the risk.

Line 93-102 - A table would be helpful to summarise the IUCN threat categories and how they link with assumed abundance. It is unclear what "criteria A1-4 by sub-criteria a, b, c, d, e" means, for example. I got completely lost in the remainder of this paragraph

Line 153 - In general, the authors should be cautious of saying Species with X characteristic have more zoonotic viruses. Despite trying to account for biases, there are still detection issues there, and phrasing should reflect that (as suggested in the abstract)

Line 159 - should this be "one or more of the zoonotic viruses"

Line 160 - (58.8%, n= 84)

Line 161 - quantify what is meant by 'vast majority'

Figure 1 - It is unclear which circles the lines are pointing to in B.

Line 164 - 167 - Clarify whether both viruses and hosts or just hosts were grouped by taxonomic order.

Lines 167-169 - Clarify what proportion of all terrestrial mammalian species these orders represent.

Table 1: In assessing the effect of "Least Concern increasing", it is unclear in the text whether includes or accounts for domestic species or not (lines 195-197).

Table 1: Suggest present this from top to bottom in order of importance -i.e. as outlined in lines 187-189. Also, the final model structure is not given, and it is unclear whether the effects

presented take into account the hierarchical structure of the data e.g. only threatened species are classified under “IUCN Criteria for Threatened Status”

Line 303 - 304 - The wording here is helpful to clarify that effects being reported take into account other effects, but this is not mentioned elsewhere.

Figure 2: It took me a while to see that the lines were connecting the animal silhouettes to the relevant circles for that species. A minor edit that might better highlight this would be to delete the grey lines after they pass (i.e. above) the silhouettes so it looks more like a line connecting to the silhouette.

It is also confusing to have these silhouettes overlapping with the (long) Population size reduction text. I suggest abbreviating this text to a few words in the figure, and have the full description in the Figure caption. I think it would also assist digestibility of the figure if the four categories in blue were also labelled A-D.

Lines 276 - 321 - Much of the text in this section is discussion, not results. It should be moved to the discussion.

Discussion - By this point, taking into account the statement on lines 187 - 189 and the content in table 1, the take-home message I have inferred is that the research effort is the main determinant for detection of zoonotic viruses, followed by whether the species is domestic or not.

Domesticated species contribute far more zoonoses than terrestrial mammalian wildlife species (lines 262-267). This should be the main finding, and any mention of the smaller effects of population size or IUCN status (which the authors are putting forward as the main finding) should be placed within that context. Of the fraction of zoonotic viruses that come from wildlife, significant effects of population size and threat status are evident, but how many viruses/how much risk does this represent in real terms?

Lines 334-340 - Species distribution, human distribution and habitats are not taken into account. The authors need to demonstrate the link between large population sizes and human contact, or explicitly state that this is a foundational assumption of these analyses. This is partly addressed in lines 341 - 342, but range overlap does not equate with habitat overlap

Line 405-406: I do not agree that there is evidence to support that “abundant mammal species harbor more zoonotic viruses than less abundant species”. There has been greater opportunity for those to spillover into people, however the results for the threat categories show that that certain circumstances just allow us to be aware of the zoonotic viruses present in less abundant species.

Line 409 - 410: Given the majority of zoonotic viruses come from domestic animals to humans, how much effect will this provide versus simply focussing on domestic animal health? There is no mention of the force of infection from domestic animals here.

Additionally, I accept that it is difficult to study, and outside of the scope of this manuscript, but the number of zoonotic viruses from any given species does not reflect the burden of disease in people. The need for this information should be highlighted in the discussion.

Supplementary methods

Lines 8-10 - For completeness, add a date as to when the search was conducted, and the range of years of studies returned. Also, in the search terms, were wildcards included? e.g. zoonos* (to catch zoonosis and zoonoses). Was this list cross checked against existing databases (e.g. those on bat and rodent viruses) to see whether any were missed with this approach?

Lines S28-30 - Although it is stated in the supplementary materials, clarify here that evidence included PCR, virus isolation and serological evidence

Lines S69 - It is helpful if the Supplementary methods are readable on their own - Explain what the ‘ZIP model’ is.

Author's Response to Decision Letter for (RSPB-2019-1561.R0)

See Appendix A.

RSPB-2019-2736.R0

Review form: Reviewer 1

Recommendation

Accept as is

Scientific importance: Is the manuscript an original and important contribution to its field?

Excellent

General interest: Is the paper of sufficient general interest?

Excellent

Quality of the paper: Is the overall quality of the paper suitable?

Good

Is the length of the paper justified?

Yes

Should the paper be seen by a specialist statistical reviewer?

Yes

Do you have any concerns about statistical analyses in this paper? If so, please specify them explicitly in your report.

No

It is a condition of publication that authors make their supporting data, code and materials available - either as supplementary material or hosted in an external repository. Please rate, if applicable, the supporting data on the following criteria.

Is it accessible?

Yes

Is it clear?

Yes

Is it adequate?

Yes

Do you have any ethical concerns with this paper?

No

Comments to the Author

Thank you for addressing all my comments.

Review form: Reviewer 4

Recommendation

Major revision is needed (please make suggestions in comments)

Scientific importance: Is the manuscript an original and important contribution to its field?
Excellent

General interest: Is the paper of sufficient general interest?
Excellent

Quality of the paper: Is the overall quality of the paper suitable?
Good

Is the length of the paper justified?
Yes

Should the paper be seen by a specialist statistical reviewer?
No

Do you have any concerns about statistical analyses in this paper? If so, please specify them explicitly in your report.
Yes

It is a condition of publication that authors make their supporting data, code and materials available - either as supplementary material or hosted in an external repository. Please rate, if applicable, the supporting data on the following criteria.

Is it accessible?
Yes

Is it clear?
Yes

Is it adequate?
Yes

Do you have any ethical concerns with this paper?
No

Comments to the Author

I have now read the manuscript entitled Global shifts in mammalian declines reveal key predictors of virus spillover risk by Johnson and collaborators, and submitted to Proceedings B, London. I really enjoyed reading this very nice work, which is well written and reached sound conclusions based on clear and simple results. Reading the manuscript is easy, and the work in general is based on appropriate statistical methods, i.e., general linearized mixed modeling. However, I have several concerns about the use of statistical methods and the context of their employment by the authors.

1. It is still hard to follow the statistical explanation even if the authors made an effort in this new amended version of their manuscript. I do understand that they used a forward regression procedure to reach a final model, i.e., a minimal model, but they poorly explained this in the document. I understand that the final model is minimal, best explained equation-s, but this should be clearly discussed. The authors should better explained what they really did for the reader (see following comment).

2. Associated with this first comment, forward regression modeling is highly sensitive to the introduction of the first independent variable in the model, and in particular in the case where this-ese variable-s is-are non-continuous variable-s, e.g., ordinal, categorical, dichotomous. In general it is better to use forward and backward procedures separately, and then compare and discuss results obtained, or to use stepwise regression procedures. In addition, I am not sure that permutations of independent variables have been done in this work, since forward modeling is

very sensitive to the introduction of the first variable. I would recommend the authors to do this proceeding to changing ranking order of independent variables introduced in the GLMM models, and see what happens.

3. Final, minimal models retained are zero-inflated Poisson regression models, and the authors used other types of models e.g., zero-inflated negative binomial regression models as well, and I am curious to see the differences observed between the different models. This could appear as a supplementary material. Model fitting for this category of dependent variable distribution is generally difficult, with lack or difficulty of convergence, so presenting the different results obtained could be important for the reader.

4. The discussion/conclusion section should discuss about the spatial scale of this study and the results obtained, and replace them within the framework of hierarchical systems; determinants that are observed/obtained at one hierarchical level can be different for other scales. This is true for global species abundances as predictors of zoonotic virus species (so at global scale), but local mammal species abundance can be or not correlated to global mammal species abundance and appear not to be a good predictor of local zoonotic virus diversity. The authors should better discuss this point in this section.

As a reviewer, I would appreciate that the authors may reply to the above comments, notably point 2 which is the more objectionable.

Review form: Reviewer 5

Recommendation

Major revision is needed (please make suggestions in comments)

Scientific importance: Is the manuscript an original and important contribution to its field?

Good

General interest: Is the paper of sufficient general interest?

Excellent

Quality of the paper: Is the overall quality of the paper suitable?

Good

Is the length of the paper justified?

Yes

Should the paper be seen by a specialist statistical reviewer?

Yes

Do you have any concerns about statistical analyses in this paper? If so, please specify them explicitly in your report.

Yes

It is a condition of publication that authors make their supporting data, code and materials available - either as supplementary material or hosted in an external repository. Please rate, if applicable, the supporting data on the following criteria.

Is it accessible?

Yes

Is it clear?

Yes

Is it adequate?

Yes

Do you have any ethical concerns with this paper?

No

Comments to the Author

General comments:

As other reviewers have highlighted, the article by Johnson and colleagues has involved a significant effort, from compiling a dataset to systematizing information about zoonotic viral infections across mammals. In my opinion, this article will be of broad interest for the Proc b readership and I anticipate that it could be highly cited because it establishes a link between species conservation and human health. I hope the authors find my comments useful.

The authors have done a great job responding to comments from previous reviewers. After carefully reviewing the previous version of the article and the author's responses my major concern with its current presentation comes from the lack of methodological details to allow for replication of the analyses and the presentation of some results, which require attention.

Since the major advance of this work is related to assessing the importance of conservation status and population trends of mammalian species in explaining the number of virus shared with humans without exploring mechanistic relationships between mammalian declines and virus spillover events, I find the title a bit confusing or misleading up to some point. I suggest rephrasing the title to reflect the findings of the article better.

Introduction

Lines 33 – 35. I find the phrase redundant in its terminology. Zoonotic diseases are by definition infectious diseases transmitted from animals to humans. I suggest rephrasing (e.g. "Infections that originate from animals and infect people (zoonotic diseases) comprise the majority...").

Line 40 – 41. What do the authors mean by "disease detection capabilities..."? If it refers to the difficulties to detect infectious agents in wild species I suggest rephrasing, since the concept of disease refers to an altered physiological state experienced by an organism that affects its health not to the presence of an infectious agent in the organism.

Lines 41 – 42. I suggest providing support for why spillover events in humans are "massively under-reported".

Line 51. I find the concept of "range size" in the following sentence a bit confusing: "...detections of zoonotic diseases, such as larger range size and fecundity among rodents (7),..." Do the authors refer to home range sizes or distributional range? I suggest rephrasing to clarify the concept.

Line 77. I suggest changing the verb "predict" in the beginning of the line for "related" or alike to acknowledge that the model formulations used aimed to detect statistical relationships not actual predictive capacity of the explanatory variables considered.

Materials and methods

Overall, I find the materials and methods difficult to follow. After reading the SI and the description of methods in the main text multiple times. A clearer description of the methods is crucial to allow for replication of the analysis and therefore results. It is also necessary since there is no code is associated to the manuscript for readers to replicate the analysis. In particular, the model formulations are confusing to me based on the narrative. For instance, from table S1, I understand that the whole dataset was used, however IUCN criteria for threatened species only applies to the subset of species (lines 52 – 67) with the levels of "VU, EN and CE" in the variable "Trends in Species Declines" (Table S1).

I have a major concern about how the categorization of IUCN red list status and population trend (Table 1, and named as Trends in Species declines in table S1) were incorporated in the analysis.

If I understand well, these variables were included in the models as categorical predictors ignoring their ordinal nature (e.g. commonly species would progress gradually in their categorization from the lowest to highest conservation concern (or vice versa). A potential approach to consider this is to include the conservation categories as numeric variables (and different transformations to take into account potential non-linearity) from, for instance 1 for the category with the lowest risk of extinction to 12 to the highest extinction risk (Critically endangered). In this way, the authors would explicitly test for the importance of conservation status in driving the richness of zoonotic viruses hosted by a particular species. The suggested 'treatment' of this variable is further supported by the author's statement that conservation categories are proxies of species abundances (lines 103 - 105) which is of continuous numeric nature.

Since the analytical approach is complex in the sense that it uses multiple levels of categorization of variables (lines 51 - 67 in expanded methods in SI) I suggest adding a visual diagram (e.g. in the structure of a decision tree diagram) to explain the IUCN metrics and their adaptations included in the models. If the authors decide to include a diagram, I think it would be good to highlight the variables that were used for the different steps in the analysis (e.g. multivariable regression and ZIP).

Considering the model selection approach implemented by the authors, the inclusion of the tables with the models at each step of inclusion of variables should be presented as supplementary information for better informing the readers with the behaviour of the different explanatory variables.

Lines 152 - 154: What do the authors mean by "Taxonomic order was evaluated as a clustered random effect ..."? Were the ZIP models formulated including order as a random effect? If so, please clarify whether Order was included as random effect in all model formulations and phrase the model description accordingly (For instance: Line 148, A mixed effect zero-inflated regression modelling approach was used...")

Line 79 in SI: Is Taxonomic Order treated as a random effect? Consider previous comment about lines 152 - 154.

Table S1 Overall the table organization is a bit confusing. I suggest to present the ΔAIC calculated as (AIC_{fitted} - AIC_{full}) in this way the sign of the ΔAIC would show direct relationship with "better or worse fit" as negative values would relate to lower AIC than the full model and therefore "better fit". In addition, I suggest reorganizing the rows from the most supported model (full in this case) to the least supported one.

Results

Figure 1B: the inclusion of humans in the plot makes it difficult to visualize the other groups. I suggest removing the human species from the plot. This would help communicate how the abundance of other species relates to the proportion of total zoonotic viruses they host.

Figure 2: I really liked this figure. I suggest adding the scale (circles) at the bottom of the figure to facilitate interpretation by the reader without necessity of reading the caption while interpreting the figure.

Figure 3: I find this figure difficult to follow. In its current form, it does not allow the reader to visualize the level of connection (virus shared) among groups. I suggest removing the viruses as nodes in the network and instead use the thickness of the links to represent the number of viruses connecting the host groups (homologous to number of contacts among individuals in a social network), including humans. This would help readers visualize how well connected the different host species are but also their contribution to human infections individually and as groups.

Lines 184 - 185. I find confusing the use of percentages, are they the percentage of total zoonotic viruses hosted by those groups? Check the percentage for Chiroptera "(n=30%)".

Lines 199 - 202. This sentence (if required) should be included in section "Materials and Methods".

Lines 230 - 239. A visualization of the 'dose response' relationship reported would help better report the results.

Lines 248 – 257: This section presents findings from other studies and hypothesis from the authors. I suggest moving them to the discussion section.

Lines 263 – 271. As in the previous comment, this section should be moved to the discussion since the authors discuss results and present their hypothesis about transmission of pathogens between humans and highly threatened species.

Line 273. I find this subtitle confusing. The subtitle refers to a “positive feedback” between the driver of mammalian declines and the number of viruses shared with humans, however this specific loop effect was not assessed. It is also not clear if the ‘effect’ reported in this section and table 2 (under the “IUCN Criteria for Threatened Status subsection) was tested in a sub-model that included only threatened species (?), if so, please rephrase sections accordingly.

Line 278. What do the authors mean by “given all other factors in the multivariate model”? Please rephrase the sentence for clarification.

Lines 279 – 281. As previous comment for lines 248 – 257.

Lines 283 – 286. I find the reference to figure 2 confusing. The sentence refers to the model outcomes (“predictions”) which are the coefficients, therefore I was expecting Figure 2 to show the model estimates (coefficients). However, the figure shows the ‘raw’ number of viruses shared between different taxa (by conservation status) and humans. I suggest referring to figure 2 to report the raw number of zoonotic viruses by group and maybe, include a replicate of figure 2 reporting the model estimates.

Lines 286 – 297. As the previous comment about lines 248 - 257.

Lines 346 – 361. As the previous comment, I find these lines to be material for the discussion.

Discussion

Overall, I think a more direct link between the subheadings in the results section, and the narrative in the discussion is needed. For instance, the discussion does not mention the role of primates and bats in the number of viruses shared with humans, which was presented in the results as a subheading. Please ignore this comment if lines 346 – 361 are included as discussion material.

Line 381: I find the term “community size” a bit confusing. Does this refer to population size (same species of hosts) or the actual assembly of host species? I think that the first would be more appropriate for the context (?).

Lines 397 – 403. Please add references to these statements.

Line 422: The assumption that a species is a competent host for spillover because it was seropositive is misplaced. An individual/species could be seropositive (exposed) but not a competent host (i.e. suitable for transmission), or not be of epidemiological relevance as a ‘source’ of infection. The authors must acknowledge how this assumption in the treatment of serology and PCR data could influence their inferences. One potential consequence could be an overestimation of species as a “source” of pathogens, especially in cases where multiple species may seroconvert, but only a few are main shedders. For example, in the case of Hendra virus in Australia, although *Pteropus poliocephalus* and other pteropid bats may be seropositive against the virus (supporting exposure), PCR results and viral isolation support that *P. alecto* and *P. conspicillatus* are the main reservoirs (see Field HE (2016) Hendra virus ecology and transmission, Current opinion in Virology, 16:120-125, and references). Based on this case (Hendra virus), the list of hosts presented in the supplementary information seems far-reaching and unsupported.

Line 431 - 432. I suggest adding a table similar to table S1, including the model formulations with and without the number of publications as variable. In this way, the authors would support the statement about the explanatory importance of the different variables being consistent.

Line 432. This line refers to the statistical behaviour of variables in the model formulations, however the reference included (18) is inappropriate. The reference is to the Red List of the IUCN.

Lines 437 – 439. Could the authors make some suggestions or recommendations for how “investigator-driven” studies could build better data for understanding global patterns? It would also be useful to add an opinion or view about what aspects, processes or inferences could be gained from continuous updates or re-evaluation of models.

References

References 8, 9, 41 are not referred to in the text.

Decision letter (RSPB-2019-2736.R0)

10-Jan-2020

Dear Dr Johnson:

Your manuscript has now been peer reviewed and the reviews have been assessed by an Associate Editor. The reviewers' comments (not including confidential comments to the Editor) are included at the end of this email for your reference. As you will see, the reviewers have raised some concerns with your manuscript and we would like to invite you to revise your manuscript to address them.

Research ethics:

Use of animals and field studies:

Please submit a copy of your revised paper within three weeks. If we do not hear from you within this time your manuscript will be rejected. If you are unable to meet this deadline please let us know as soon as possible, as we may be able to grant a short extension.

Best wishes,
Professor Hans Heesterbeek
mailto: proceedingsb@royalsociety.org

Reviewer(s)' Comments to Author:

Referee: 1

Comments to the Author(s).
Thank you for addressing all my comments.

Referee: 4

Comments to the Author(s).

I have now read the manuscript entitled Global shifts in mammalian declines reveal key predictors of

virus spillover risk by Johnson and collaborators, and submitted to Proceedings B, London.

I really enjoyed reading this very nice work, which is well written and reached sound conclusions based on clear and simple results. Reading the manuscript is easy, and the work in general is based on appropriate statistical methods, i.e., general linearized mixed modeling.

However, I have several concerns about the use of statistical methods and the context of their employment by the authors.

1. It is still hard to follow the statistical explanation even if the authors made an effort in this new amended version of their manuscript. I do understand that they used a forward regression procedure to reach a final model, i.e., a minimal model, but they poorly explained this in the document. I understand that the final model is minimal, best explained equation-s, but this should be clearly discussed. The authors should better explained what they really did for the reader (see following comment).

2. Associated with this first comment, forward regression modeling is highly sensitive to the introduction of the first independent variable in the model, and in particular in the case where this-ese variable-s is-are non-continuous variable-s, e.g., ordinal, categorical, dichotomous. In general it is better to use forward and backward procedures separately, and then compare and discuss results obtained, or to use stepwise regression procedures. In addition, I am not sure that permutations of independent variables have been done in this work, since forward modeling is very sensitive to the introduction of the first variable. I would recommend the authors to do this proceeding to changing ranking order of independent variables introduced in the GLMM models, and see what happens.

3. Final, minimal models retained are zero-inflated Poisson regression models, and the authors used other types of models e.g., zero-inflated negative binomial regression models as well, and I am curious to see the differences observed between the different models. This could appear as a supplementary material. Model fitting for this category of dependent variable distribution is generally difficult, with lack or difficulty of convergence, so presenting the different results obtained could be important for the reader.

4. The discussion/conclusion section should discuss about the spatial scale of this study and the results obtained, and replace them within the framework of hierarchical systems; determinants that are observed/obtained at one hierarchical level can be different for other scales. This is true for global species abundances as predictors of zoonotic virus species (so at global scale), but local mammal species abundance can be or not correlated to global mammal species abundance and appear not to be a good predictor of local zoonotic virus diversity. The authors should better discuss this point in this section.

As a reviewer, I would appreciate that the authors may reply to the above comments, notably point 2 which is the more objectionable.

Referee: 5

Comments to the Author(s).

General comments:

As other reviewers have highlighted, the article by Johnson and colleagues has involved a significant effort, from compiling a dataset to systematizing information about zoonotic viral infections across mammals. In my opinion, this article will be of broad interest for the Proc B readership and I anticipate that it could be highly cited because it establishes a link between species conservation and human health. I hope the authors find my comments useful.

The authors have done a great job responding to comments from previous reviewers. After carefully reviewing the previous version of the article and the author's responses my major

concern with its current presentation comes from the lack of methodological details to allow for replication of the analyses and the presentation of some results, which require attention. Since the major advance of this work is related to assessing the importance of conservation status and population trends of mammalian species in explaining the number of virus shared with humans without exploring mechanistic relationships between mammalian declines and virus spillover events, I find the title a bit confusing or misleading up to some point. I suggest rephrasing the title to reflect the findings of the article better.

Introduction

Lines 33 – 35. I find the phrase redundant in its terminology. Zoonotic diseases are by definition infectious diseases transmitted from animals to humans. I suggest rephrasing (e.g. “Infections that originate from animals and infect people (zoonotic diseases) comprise the majority...”).

Line 40 – 41. What do the authors mean by “disease detection capabilities...”? If it refers to the difficulties to detect infectious agents in wild species I suggest rephrasing, since the concept of disease refers to an altered physiological state experienced by an organism that affects its health not to the presence of an infectious agent in the organism.

Lines 41 – 42. I suggest providing support for why spillover events in humans are “massively under-reported”.

Line 51. I find the concept of “range size” in the following sentence a bit confusing: “...detections of zoonotic diseases, such as larger range size and fecundity among rodents (7),...” Do the authors refer to home range sizes or distributional range? I suggest rephrasing to clarify the concept.

Line 77. I suggest changing the verb “predict” in the beginning of the line for “related” or alike to acknowledge that the model formulations used aimed to detect statistical relationships not actual predictive capacity of the explanatory variables considered.

Materials and methods

Overall, I find the materials and methods difficult to follow. After reading the SI and the description of methods in the main text multiple times. A clearer description of the methods is crucial to allow for replication of the analysis and therefore results. It is also necessary since there is no code is associated to the manuscript for readers to replicate the analysis. In particular, the model formulations are confusing to me based on the narrative. For instance, from table S1, I understand that the whole dataset was used, however IUCN criteria for threatened species only applies to the subset of species (lines 52 – 67) with the levels of “VU, EN and CE” in the variable “Trends in Species Declines” (Table S1).

I have a major concern about how the categorization of IUCN red list status and population trend (Table 1, and named as Trends in Species declines in table S1) were incorporated in the analysis. If I understand well, these variables were included in the models as categorical predictors ignoring their ordinal nature (e.g. commonly species would progress gradually in their categorization from the lowest to highest conservation concern (or vice versa). A potential approach to consider this is to include the conservation categories as numeric variables (and different transformations to take into account potential non-linearity) from, for instance 1 for the category with the lowest risk of extinction to 12 to the highest extinction risk (Critically endangered). In this way, the authors would explicitly test for the importance of conservation status in driving the richness of zoonotic viruses hosted by a particular species. The suggested ‘treatment’ of this variable is further supported by the author’s statement that conservation categories are proxies of species abundances (lines 103 – 105) which is of continuous numeric nature.

Since the analytical approach is complex in the sense that it uses multiple levels of categorization of variables (lines 51 – 67 in expanded methods in SI) I suggest adding a visual diagram (e.g. in the structure of a decision tree diagram) to explain the IUCN metrics and their adaptations included in the models. If the authors decide to include a diagram, I think it would be good to highlight the variables that were used for the different steps in the analysis (e.g. multivariable regression and ZIP).

Considering the model selection approach implemented by the authors, the inclusion of the tables with the models at each step of inclusion of variables should be presented as supplementary information for better informing the readers with the behaviour of the different explanatory variables.

Lines 152 – 154: What do the authors mean by “Taxonomic order was evaluated as a clustered random effect ...” ? Were the ZIP models formulated including order as a random effect? If so, please clarify whether Order was included as random effect in all model formulations and phrase the model description accordingly (For instance: Line 148, A mixed effect zero-inflated regression modelling approach was used...”

Line 79 in SI: Is Taxonomic Order treated as a random effect? Consider previous comment about lines 152 – 154.

Table S1 Overall the table organization is a bit confusing. I suggest to present the Δ AIC calculated as (AIC_{fitted} – AIC_{full}) in this way the sign of the Δ AIC would show direct relationship with “better or worse fit” as negative values would relate to lower AIC than the full model and therefore “better fit”. In addition, I suggest reorganizing the rows from the most supported model (full in this case) to the least supported one.

Results

Figure 1B: the inclusion of humans in the plot makes it difficult to visualize the other groups. I suggest removing the human species from the plot. This would help communicate how the abundance of other species relates to the proportion of total zoonotic viruses they host.

Figure 2: I really liked this figure. I suggest adding the scale (circles) at the bottom of the figure to facilitate interpretation by the reader without necessity of reading the caption while interpreting the figure.

Figure 3: I find this figure difficult to follow. In its current form, it does not allow the reader to visualize the level of connection (virus shared) among groups. I suggest removing the viruses as nodes in the network and instead use the thickness of the links to represent the number of viruses connecting the host groups (homologous to number of contacts among individuals in a social network), including humans. This would help readers visualize how well connected the different host species are but also their contribution to human infections individually and as groups.

Lines 184 – 185. I find confusing the use of percentages, are they the percentage of total zoonotic viruses hosted by those groups? Check the percentage for Chiroptera “(n=30%)”.

Lines 199 – 202. This sentence (if required) should be included in section “Materials and Methods”.

Lines 230 – 239. A visualization of the ‘dose response’ relationship reported would help better report the results.

Lines 248 – 257: This section presents findings from other studies and hypothesis from the authors. I suggest moving them to the discussion section.

Lines 263 – 271. As in the previous comment, this section should be moved to the discussion since the authors discuss results and present their hypothesis about transmission of pathogens between humans and highly threatened species.

Line 273. I find this subtitle confusing. The subtitle refers to a “positive feedback” between the driver of mammalian declines and the number of viruses shared with humans, however this specific loop effect was not assessed. It is also not clear if the ‘effect’ reported in this section and table 2 (under the “IUCN Criteria for Threatened Status subsection) was tested in a sub-model that included only threatened species (?), if so, please rephrase sections accordingly.

Line 278. What do the authors mean by “given all other factors in the multivariate model”? Please rephrase the sentence for clarification.

Lines 279 – 281. As previous comment for lines 248 – 257.

Lines 283 – 286. I find the reference to figure 2 confusing. The sentence refers to the model outcomes (“predictions”) which are the coefficients, therefore I was expecting Figure 2 to show the model estimates (coefficients). However, the figure shows the ‘raw’ number of viruses shared between different taxa (by conservation status) and humans. I suggest referring to figure 2 to

report the raw number of zoonotic viruses by group and maybe, include a replicate of figure 2 reporting the model estimates.

Lines 286 – 297. As the previous comment about lines 248 - 257.

Lines 346 – 361. As the previous comment, I find these lines to be material for the discussion.

Discussion

Overall, I think a more direct link between the subheadings in the results section, and the narrative in the discussion is needed. For instance, the discussion does not mention the role of primates and bats in the number of viruses shared with humans, which was presented in the results as a subheading. Please ignore this comment if lines 346 – 361 are included as discussion material.

Line 381: I find the term “community size” a bit confusing. Does this refer to population size (same species of hosts) or the actual assembly of host species? I think that the first would be more appropriate for the context (?).

Lines 397 – 403. Please add references to these statements.

Line 422: The assumption that a species is a competent host for spillover because it was seropositive is misplaced. An individual/species could be seropositive (exposed) but not a competent host (i.e. suitable for transmission), or not be of epidemiological relevance as a ‘source’ of infection. The authors must acknowledge how this assumption in the treatment of serology and PCR data could influence their inferences. One potential consequence could be an overestimation of species as a “source” of pathogens, especially in cases where multiple species may seroconvert, but only a few are main shedders. For example, in the case of Hendra virus in Australia, although *Pteropus poliocephalus* and other pteropid bats may be seropositive against the virus (supporting exposure), PCR results and viral isolation support that *P. alecto* and *P. conspicillatus* are the main reservoirs (see Field HE (2016) Hendra virus ecology and transmission, *Current opinion in Virology*, 16:120-125, and references). Based on this case (Hendra virus), the list of hosts presented in the supplementary information seems far-reaching and unsupported.

Line 431 - 432. I suggest adding a table similar to table S1, including the model formulations with and without the number of publications as variable. In this way, the authors would support the statement about the explanatory importance of the different variables being consistent.

Line 432. This line refers to the statistical behaviour of variables in the model formulations, however the reference included (18) is inappropriate. The reference is to the Red List of the IUCN.

Lines 437 – 439. Could the authors make some suggestions or recommendations for how “investigator-driven” studies could build better data for understanding global patterns? It would also be useful to add an opinion or view about what aspects, processes or inferences could be gained from continuous updates or re-evaluation of models.

References

References 8, 9, 41 are not referred to in the text.

Author's Response to Decision Letter for (RSPB-2019-2736.R0)

See Appendix B.

RSPB-2019-2736.R1 (Revision)

Review form: Reviewer 1

Recommendation

Accept as is

Scientific importance: Is the manuscript an original and important contribution to its field?

Excellent

General interest: Is the paper of sufficient general interest?

Excellent

Quality of the paper: Is the overall quality of the paper suitable?

Good

Is the length of the paper justified?

Yes

Should the paper be seen by a specialist statistical reviewer?

Yes

Do you have any concerns about statistical analyses in this paper? If so, please specify them explicitly in your report.

No

It is a condition of publication that authors make their supporting data, code and materials available - either as supplementary material or hosted in an external repository. Please rate, if applicable, the supporting data on the following criteria.

Is it accessible?

Yes

Is it clear?

Yes

Is it adequate?

Yes

Do you have any ethical concerns with this paper?

No

Comments to the Author

I have no further comments, but see other reviewers did.

Review form: Reviewer 5

Recommendation

Accept with minor revision (please list in comments)

Scientific importance: Is the manuscript an original and important contribution to its field?
Good

General interest: Is the paper of sufficient general interest?
Good

Quality of the paper: Is the overall quality of the paper suitable?
Good

Is the length of the paper justified?
Yes

Should the paper be seen by a specialist statistical reviewer?
No

Do you have any concerns about statistical analyses in this paper? If so, please specify them explicitly in your report.
No

It is a condition of publication that authors make their supporting data, code and materials available - either as supplementary material or hosted in an external repository. Please rate, if applicable, the supporting data on the following criteria.

Is it accessible?
Yes

Is it clear?
Yes

Is it adequate?
Yes

Do you have any ethical concerns with this paper?
No

Comments to the Author

I appreciate the authors' consideration of my comments. I am glad they found my comments helpful. Thank you for your responses and revisions.

Some minor comments:

Line 38, 246, 256, 420: Please replace "disease" by "pathogen".

Line 43: remove "and" after "between".

Line 150, 323, 419: replace "to date" by "to the date of the study" or similar to acknowledge the time difference between the data used and publication date.

Line 419: please replace "zoonotic diseases" by "zoonotic pathogens" since the study focuses on the detection of pathogens or exposure of host species to them, not the clinical manifestation of the infections.

Line 424: Please rephrase "Surveillance of people..." to clarify that the authors suggest assessing or monitoring of health in those people and not security surveillance.

Decision letter (RSPB-2019-2736.R1)

05-Mar-2020

Dear Dr Johnson

I am pleased to inform you that your manuscript RSPB-2019-2736.R1 entitled "Global shifts in mammalian population trends reveal key predictors of virus spillover risk" has been accepted for publication in Proceedings B.

The referees have recommended publication, but also suggest some minor revisions to your manuscript. Therefore, I invite you to respond to the comments and revise your manuscript. Because the schedule for publication is very tight, it is a condition of publication that you submit the revised version of your manuscript within 7 days. If you do not think you will be able to meet this date please let us know.

Sincerely,

Professor Hans Heesterbeek

Reviewer(s)' Comments to Author:

Referee: 1

Comments to the Author(s)

I have no further comments, but see other reviewers did.

Referee: 5

Comments to the Author(s)

I appreciate the authors' consideration of my comments. I am glad they found my comments helpful. Thank you for your responses and revisions.

Some minor comments:

Line 38, 246, 256, 420: Please replace "disease" by "pathogen".

Line 43: remove "and" after "between".

Line 150, 323, 419: replace "to date" by "to the date of the study" or similar to acknowledge the time difference between the data used and publication date.

Line 419: please replace "zoonotic diseases" by "zoonotic pathogens" since the study focuses on the detection of pathogens or exposure of host species to them, not the clinical manifestation of the infections.

Line 424: Please rephrase "Surveillance of people..." to clarify that the authors suggest assessing or monitoring of health in those people and not security surveillance.

Decision letter (RSPB-2019-2736.R2)

13-Mar-2020

Dear Dr Johnson

I am pleased to inform you that your manuscript entitled "Global shifts in mammalian population trends reveal key predictors of virus spillover risk" has been accepted for publication in Proceedings B.

Open Access

Paper charges

You are allowed to post any version of your manuscript on a personal website, repository or

preprint server. However, the work remains under media embargo and you should not discuss it with the press until the date of publication. Please visit <https://royalsociety.org/journals/ethics-policies/media-embargo> for more information.

Sincerely,
Editor, Proceedings B
<mailto:proceedingsb@royalsociety.org>

Appendix A

UNIVERSITY OF CALIFORNIA, DAVIS

BERKELEY • DAVIS • IRVINE • LOS ANGELES • MERCED • RIVERSIDE • SAN DIEGO • SAN FRANCISCO

SANTA BARBARA • SANTA CRUZ

SCHOOL OF VETERINARY MEDICINE
WILDLIFE HEALTH CENTER
UNIVERSITY OF CALIFORNIA
(530) 752-4167
FAX (530) 752-3318
<http://www.vetmed.ucdavis.edu/whc>

ONE SHIELDS AVENUE
DAVIS, CALIFORNIA 95616-8734

November 21, 2019

Dear Professor Heersterbeek,

Please find a revised version of our manuscript RSPB-2019-1561 entitled "Global shifts in mammalian declines reveal key predictors of virus spillover risk" for consideration by *Proceeding of the Royal Society B: Biological Sciences*. We appreciate the expert reviews and suggestions to improve the manuscript. We have made recommended edits to the main text and Supplemental Information, updated the data and models as suggested, and improved scientific clarity in presentation of the research. We think the revisions made in light of the reviews by the referees have substantially improved this paper. Please find below a point by point response to all reviewers' comments (*in blue italics*).

Do not hesitate to contact us if we can provide additional information on revisions to this paper. Thank you for the opportunity to address suggestions and re-submit this manuscript for consideration.

Sincerely,

Christine K. Johnson
University of California, Davis
ckjohnson@ucdavis.edu

Associate Editor Comments

Associate Editor

Board Member: 1

Comments to Author:

It's clear all referees found the manuscript interesting, and that it contributes valuable results to an important field- a view which I concur with also. There are many minor points raised by the referees, but focusing on the more key points:

Referee 1 raises the issue that perhaps some viruses included are only potentially zoonotic and this should be checked.

We have re-reviewed the evidence in the literature and have concluded that, based on our inclusion criteria, 4 viruses with only potential zoonotic status as noted should be excluded. Because viruses were removed from the data, analyses were redone, and the text and figures updated accordingly. Our statistical results and findings from the regression model remained largely the same.

Referee 1 also raises that main text statistical methods are not easy to follow, but that the SI is necessary for the reader to get a grip on what was actually done.

We have revised the methods section for clarity by moving information on our statistical approach from the Supplementary Materials into the main paper as suggested by Reviewer 1.

Ref 3 raises a number of points, but they pertain mainly to the presentation of the results, and deductions that follow from these. There are a lot of specific points drawn out for clarification, but see the comments on figure 2 in particular.

Changes to figure 2 were made as suggested below.

A few minor points on the figures from me:

- Fig 1: what is the area of the circles representing?

Clarifications for Fig 1 area of circles were added to the figure legends. Area of the circles in Fig 1a is relative to the proportion of zoonotic viruses among species in that order out of the total number of zoonotic viruses found in all mammalian species. In Fig 1B, the area of the circles reflects the relative abundance of that species out of total abundance for domesticated species and humans.

- Fig 2: when it says "size" of circles for this figure, is it area? Also give some scale for these.

Correct, we mean area, and we have included the range of scale in the figure legend.

- Fig 2: I agree with ref 3 that it took a while to see what's going on here. In addition to ref 3's suggestions, eliminate the extra grey line at the left, reduce the text through the figure. Consider making the grey lines match to the species colours. Consider putting the species icons on the same row (make the text vertical).

Revisions to Fig 2 were made to reflect all suggestions, including eliminating grey lines, having lines match species color, putting species icons on the same row and reducing text in the figure.

- Fig 2: what is going on with perissodactyla?? They look a bit outlier and this isn't referred to in main text.

Agreed, domestic perissodactyla are outliers in terms of the high number of zoonotic viruses, and this was added to the results section.

Bringing together the views of the referees, it's clear the main text could be tightened up in places, and some of the key methods from SI could be brought in to make more clear what was actually done. Referee 2 does comment that the authors have been cautious in not over-interpreting findings, but perhaps the interpretation could be reduced or streamlined (there is some repetition).

Referee 1 Comments

Comments to the Author(s)

Overall, I enjoyed reading this work, titled “Global shifts mammalian declines reveal key predictors of virus spillover risk” by Johnson and colleagues. Personally, I feel this work addresses an important problem, the authors introduce the problem well, use established methods, present their findings clearly, and discuss the weaknesses of the work fairly. I was a little skeptical as to how they would use the IUCN data and how ‘novel’ their framework was [see abstract], as these data have significant weaknesses. However, I thought the authors used them well and quite imaginatively to come to conclusions that had until recently largely been discussed, but not rigorously analysed in the literature.

Major issues

Data:

The methods are fair and well explained and overall I agree with the decisions made and/or the authors have given sound justification for their decisions.

However, there is one potential exception, and that is that several viruses I see cited are not actually known to be zoonotic, but are potentially zoonotic. For example, the bat lyssaviruses Lagos bat virus, Shimoni bat virus, West Caucasian bat virus and Aravan virus have not, as far as I am aware, been isolated from people. Most of the others have, but for accuracy these either need excluding or the authors need to justify why they are included. My guess is that excluding these won't change the results, but you should check.

We have re-reviewed the evidence in the literature and have concluded that, based on our inclusion criteria, these 4 viruses with only potential zoonotic status as noted should be excluded from analyses because there is insufficient evidence to classify these four viruses as zoonotic as noted by this reviewer. These four viruses were thus dropped from the dataset, and all analyses and modeling procedure were redone. All of the model procedures were resilient to this update to underlying data and major findings remain unchanged, except for the taxonomic order carnivora which was dropped from the final full model ($P > 0.3$). Table 2 was updated with the final model and the text was revised accordingly. All figures were also revised to reflect this change.

After reading all the supplementary information, I believe that the methods are appropriate, but it was hard to follow. It wasn't really until the Statistical analysis section in the SI methods (starting on line 68) that I really understood how the authors had reached the modeling approach that they presented. In the details below I make some suggestions, but overall, I would prefer more of the statistical analyses text to be in the main text, particularly relating to the ZIP regression model selection and choice. This seems essential for the reader.

We revised as detailed below.

Methods

IUCN data - lines 99 to 119: I know this section is key to this, but I found this quite hard to follow. I wonder if some of this can be moved to the SI and a simpler version (perhaps an adaptation of SI Table S2) included in the main text?

We substantially shortened the description of IUCN Red List Criteria referred to above and a more detailed description of IUCN criteria was included in the Supplementary Materials. Also, Table S2 in the SI was moved to the main manuscript text as Table 1.

Statistical analyses - I preferred the presentation of the methods in the SI. Key factors that lead the authors to using this regression model are not included here and I think that they are important. Also, it was really clear to me until the results, line 159 why a zero inflated model was even being used on my first read, and that non-zoonotic virus hosts were included.

To improve clarity in description of our approach, aspects of the methods relevant to the modeling and all of the statistical methods were moved from the SI methods to the main article text methods section.

Results

Line 185-9 - this seems a crucial result, perhaps present the model structure in the main text?
This result was better defined within the context of other results by revising this sentence and referring to Table S1 for model selection and Table 2 for model presentation, as follows
“Adjusting for reporting bias prior to interpretation of other putative factors was essential, given publication of zoonotic hosts in the literature was the basis for inclusion in this study. Inclusion of number of PubMed publications improved model fit, as evidenced by change in AIC (Table S1), such that all other species-level factors could then be evaluated for their relationship with frequency of virus sharing with humans. The final zero-inflated Poisson model showed that, after accounting for reporting bias, trends in species declines, and several criteria for species reductions, taxonomic order, and domesticated species status were significantly related to the number of zoonotic viruses detected in each mammalian species (Table 2)”. Text to describe the zero-inflated structure and the full model is shown in Table S1. The final model variables, categories for each variable, and all estimated coefficients (presented as IRR) are reported in Table 2.

Line 253: "...activities." In the other sections these comments are always followed by references, but not here. I would recommend the following:

Hahn, et al (2014) "Roosting behaviour and habitat selection of Pteropus giganteus reveal potential links to Nipah virus epidemiology" J Appl Ecol, 51: 376-387

Hahn, et al (2014) "The Role of Landscape Composition and Configuration on Pteropus giganteus Roosting Ecology and Nipah Virus Spillover Risk in Bangladesh." Am J Trop Med Hyg, 90(2): 247 - 255.

Rulli, et al (2017) "The nexus between forest fragmentation in Africa and Ebola virus disease outbreaks." Scientific reports 7 (2017): 41613.

We appreciate these suggestions; we've added the latter 2 references to reflect broader evidence for our interpretation of our findings and highlight these examples of landscape change and spillover risk. We have also edited the relevant text to reflect content of these references accordingly.

Lines 327 & 377: In both instances, but especially 377 I think the work by Wilkinson et al is relevant, because they develop an explicit model of habitat fragmentation and infectious disease emergence risk:

Wilkinson, et al (2018) "Habitat fragmentation, biodiversity loss and the risk of novel infectious disease emergence" J Roy Soc Interface 15 (149), 20180403

This section was revised to include the ideas presented in this paper and add this reference.

Lines 334 - 346: here, critical community size is alluded to, but never really mentioned. The focus appears to be on population density, but some (maybe most?) wildlife species don't necessarily increase their density in the same spatial location with increased abundances, as my guess is that many species have behavioural traits (including nutrition availability and competition) that limit species densities. Maybe refer explicitly to critical community sizes, whilst also acknowledging that other transmission processes exist (e.g. frequency dependent transmission), and see: Lloyd-Smith, J.O. et al. (2005) "Should we expect population thresholds for wildlife disease?" Trends in Ecology & Evolution 20: 511-519

All excellent points; we have added specific reference to critical community size, revised text to remove emphasis on population density, and cite this paper.

SI Line 31-32: serology is more sensitive, but not in 'virus detection', as typically for acute infectious diseases this is historic infection. I recognise this is not true for some persistent infections and used for rodent arenaviruses (for example), but please correct this.

Agreed, revised as suggested.

Referee: 2

Comments to the Author(s)

Johnson et al present an original statistical analysis of conservation status (IUCN database) and carriage of 143 zoonotic viruses in 5,438 wild mammalian species. Overall, more endangered species tend to have fewer zoonotic viruses, and wild mammals have fewer zoonotic viruses than domesticated ones. Among threatened species, those that suffer from habitat loss or direct exploitation have more zoonotic viruses than others.

These results are interesting and important as they cast light on the complex relationship between two facets of the human-wildlife relationship: conservation and zoonotic spillover. These two issues are of major concern globally and have been in direct conflict in several occasions in recent years, when zoonotic spillover from wildlife has triggered culling campaigns opposed by conservationists. As illustrated by the controversies around flying foxes and Hendra virus in Australia, badgers and bovine TB in Great Britain, or bats and rabies in South America, to cite but a few, cultural and sociological factors exacerbate these conflicts and prevent their resolution, with both sides cherry-picking the scientific evidence and economic factors that align with their beliefs. What this study brings is a dispassionate and global analysis of the relationship that exist between the occurrence of zoonotic viruses and conservation status. The inclusion of sources of threat enables the authors to propose a mechanistic interpretation of the statistical association. Even though precise causation and reliable solutions are beyond the scope of this analysis, the results presented here will be very valuable to inform the debate and guide more detailed investigations of the processes underlying zoonotic spillover and wildlife population variations.

I have no major issues with this manuscript. Although I'm not qualified to properly assess the statistical analyses carried out by the authors, I know from previous experience that this type of analysis is always imperfect and comes with caveats, especially in relation to the various flaws and biases in the databases. I think the authors are quite open about these issues and have been cautious in not over-interpreting their findings.

Specific comments:

- L. 40-42: I found this sentence a bit awkward ("viruses have been linked to pandemic properties") and I'm not sure it's needed.

This sentence was removed.

- L.70-73: the proxy argument seems a bit tenuous here without further explanation. You don't need to offer this proxy to justify or motivate the analysis. I would suggest rephrasing this sentence.

This sentence was removed and these characteristics of the IUCN data were included in a revised sentence "For the majority of threatened mammal species, these IUCN metrics provide valuable context for animal-human interactions that have could have played a role in species declines."

- L.87: why only until December 2013? Given the recent increase in zoonotic virus research (not least from the PREDICT initiative), I suspect that the data may have increased substantially in the last 5 years.

We initiated data collection in December 2013 and then proceeded to analyze data and write this manuscript. We understand that this is a long delay between data collection and publication, and we appreciate the value of updating the dataset, but, unfortunately, we are not able to support the scope of work needed to verify new reports of species hosts for all zoonotic viruses. We have confirmed that this limitation is featured prominently in our methods in the main text.

- L.133: is that a fair assumption, given that zoonotic viruses are often reported in very small numbers of wild animals within species? Could this be tested in some way? Given the authors' involvement in one of the largest wildlife screening initiatives to date (PREDICT), I expect they would have access to some form of evidence for or against this claim.

Because we expected reporting bias to be an important factor in ascertaining number of zoonotic viruses, we thought best to have one measure of reporting bias that was reflective of uncertainty in the data source used to categorize species by population trend. We also established an independent measure of data deficiency at the species level by including a term for the number of research publications available in PubMed. We found that both terms were needed to adjust for detection of zoonotic viruses in a species in multivariable modeling. Data from PREDICT supports the premise that very high search effort is needed to detect viruses in target virus families, but data are unfortunately not comparable to the zoonotic virus dataset or adequate with respect to data deficient species to enable direct testing of this assumption.

- L. 266: is that a number of viral species? More generally, what taxonomic unit was used to count zoonotic viruses?

This result was clarified in the text here to indicate we mean virus species as listed in Data file S1.

Referee: 3

Comments to the Author(s)

Johnson et al undertakes to assess links between mammalian abundance and IUCN threat category with the detection of zoonotic viruses.

This work includes a significant and valuable effort to create a dataset on the known viruses detected in mammals. I am aware that this has been done previously for rodents, bats and smaller groups of taxa, but if this is really the first database of its kind for mammals, then it should be clarified further within the manuscript and celebrated as a contribution in its own right. If it builds off or expands previous similar datasets (perhaps Han et al, as is mentioned in the results), then this should also be clarified.

We collected data for this paper specifically and did not expand from other published datasets. Our methods for data collection in the literature were similar to several studies that have since been published and we thus compare our results to results in those studies. Our hope is that the zoonotic virus data provided with this study will be useful in facilitating other investigations into species predilections for hosting viruses.

The analyses rest on the assumption that IUCN species population metrics and threat measures are a proxy for the likelihood of animal- human interactions. The manuscript feels like the analysis have been run without any particular hypotheses of how these interactions might be occurring under different circumstances e.g. common but rapidly declining species vs rarer but stable species. Other presumably important factors are not considered, for example, how habitat type and habitat/dietary specialisation might correlate with human contact. Similarly, urbanisation favours survival of generalist over specialist species, and this is likely reflected in human contacts (more contacts for generalist species that thrive in urban environments), even if their overall population metrics and threat measures remain the same. The focus of the text seems to switch between which groups of species “harbour the most zoonotic viruses” versus which are of greater spillover risk, and I argue that these are not the same things. Detectability and contact rates with humans will both play a significant role.

We agree and revisions were made accordingly. Specifically, consideration of habitat and diet specialization were included in the discussion as follows “ Many of these species have habitat and dietary niches that overlap with human dwellings and agriculture, further enabling direct and indirect contact with similarly adapted sympatric species, domesticated species, and humans. “ Specific mention of generalist species in context of increased human contact was included in the discussion with a new reference as follows “Species that have increased in abundance and even expanded their range despite large scale anthropogenically driven landscape change and urbanization (36), are more likely to be generalist species that have adapted to human-dominated landscapes, Specifically, we added We also revised text to ensure consistency in reference to species harboring zoonotic viruses. We thus minimized reference to spillover risk, except when generalizing findings to inferences we are prepared to make based on the study.

I feel that the analyses are of considerable interest, however I am not currently convinced by the implementation and interpretation of the statistical analyses. In describing the analyses, the relative importance of each of the categories is unclear and the final model is not reported. For example, with most of the explanatory power consumed by research effort and domestication status, how much real contribution to zoonotic virus risk are the IUCN categories and trends actually contributing? Much of the discussion focusses on this rather than the model terms with higher importance in the model.

We highlight the final model and the relative effect of each variable on the outcome of interest in Table 2. Our model fitting procedure identified variables that significantly improved model fit and all of these variables and their categories are shown in Table 2, along with a statistic that indicates overall fit to the data (McFadden’s $R^2 = 0.247$). We have not identified a post estimation measure that could evaluate explanatory power of each variable following zero-inflated poisson regression. Thus we instead assessed relative improvement of each variable in fitting the final model using delta AIC as reported in the Supplemental Information Table S1. The change in AIC (ΔAIC) is shown for removal of each variable group from the best-fitting (full) model, which indicates the relative amount of information gained by inclusion of each variable with penalty for additional terms. Research effort certainly needed to be adjusted for in our analysis, especially to explain the very high number of species without any virus detections to date. However, the main focus of our paper is to evaluate epidemiological factors related to the variability in number of viruses among species.

Specific comments:

Line 18 - We don't know how many zoonotic viruses are in mammalian species. Change to 'detected in'. Similarly for line 20 - 'have more viruses detected'

Sentences revised as suggested.

Line 21 - A sentence is required inserted in line 21 that describes the proposed mechanisms. Also, there is no mention of domestic animals or research effort in the abstract.

The abstract was revised to include mention of domestic animals, adjustment for reporting bias, and proposed mechanisms.

Line 28 - Clarify whether the first half of this sentence is referring only to humans. I don't think that zoonotic diseases comprise the majority of emerging infectious disease threats to wildlife (consider Bd in amphibians, WNS in North American bats, DFTD in Tasmanian devils, Avian Malaria, Canine Distemper etc).

Correct; revised to indicate we refer to human threats only.

Lines 28- 55: This is a long paragraph with lots of ideas that don't easily flow. I feel that it could be edited to be more concise and more quickly get to the point. e.g. lines 32-38.

Also, to attract the broader readership that is expected for Proc B, I suggest including a few words to define zoonotic.

This paragraph was edited substantially to delete redundant text and ideas that were tangentially related to study objectives. We added a clarifying phrase to define zoonotic disease in the first sentence of the paper.

Lines 38-42 - is the focus of the paper on spillover or pandemic potential? There is no further mention of pandemics through the text and the focus on pandemic potential seems irrelevant. I suggest either rephrasing in terms of spillover potential or deleting.

Mention of pandemics was removed as suggested.

Lines 49 - 51 - clarify whether high or low body mass or human population density increases/decreases propensity to share viruses with humans

This text was clarified as suggested.

Line 51 - high risk as determined by what metric? This is a bit confusing. Suggest deleting "Within specific high risk taxa" from sentence.

Edited as suggested.

Line 52 - Replace 'numbers' with 'detections' or similar. We have no idea how many viruses most species have - can only judge what has been detected.

Replaced specified text with 'detections' as recommended.

Lines 71-73 - As discussed in general comments above, this assumption requires further justification from an ecological perspective.

This specific sentence was removed per reviewer 2.

Lines 73- 74 Provide an example here of the types of contact being referred to

This sentence in the Introduction was clarified and examples were included as follows “For the many threatened mammal species, these IUCN metrics provide valuable context for large-scale anthropogenic activities implicated in species declines (e.g., decline in habitat quality for a species), and specific animal-human contact (e.g., exploitation of a species)”.

Line 75 - this is the first time that the focus has shifted to mammals. Provide justification for this. Also clarify here that the focus is on terrestrial species

We included justification for a focus on mammals, including a justification that classifies transmission of zoonotic disease from birds as being primarily vector-borne, as follows “Wild terrestrial mammals were more likely to transmit disease to humans via direct or indirect contact, while transmission involving birds was most likely to involve vectors (8)”

Line 80-81- the results of the study don't have implications for the the risk of spillover per se - it has implications for understanding the risk, or for managing the risk.

We agree, we revised this sentence to indicate the results have . . .” implications for understanding and managing virus spillover risk”.

Line 93-102 - A table would be helpful to summarise the IUCN threat categories and how they link with assumed abundance. It is unclear what “criteria A1-4 by sub-criteria a, b, c, d, e” means, for example. I got completely lost in the remainder of this paragraph

The revised Figure 2 briefly summarizes IUCN threat categories and reference to this figure was added in the methods section. This section of the methods was substantially simplified in the main text and a more detailed description of IUCN categories is now included in the supplemental information.

Line 153 - In general, the authors should be cautious of saying Species with X characteristic have more zoonotic viruses. Despite trying to account for biases, there are still detection issues there, and phrasing should reflect that (as suggested in the abstract)

To address this concern, we clarified these statements by noting that we found these patterns after adjusting for detection biases and that all findings are based on the multi-variate model shown in Table 2 which accounts for data deficiency among species, and adjusts for number of publications in PubMed for each species.

Line 159 - should this be “one or more of the zoonotic viruses”

Correct, revised as suggested.

Line 160 - (58.8%, n= 84)

We indicate here # of mammalian species, not viruses. This was revised for clarity and n specified throughout the sentence.

Line 161 - quantify what is meant by ‘vast majority’

The specific proportion of viruses was added for each order.

Figure 1 - It is unclear which circles the lines are pointing to in B.

Figure 1b was revised so that the circle area reflects the relative abundance of each species to allow space for lines.

Line 164 - 167 - Clarify whether both viruses and hosts or just hosts were grouped by taxonomic order.

Only mammalian host species were grouped by order; this clarification was added to text.

Lines 167-169 - Clarify what proportion of all terrestrial mammalian species these orders represent.

These orders represent 72.7% of all terrestrial mammal species; this was added to text here.

Table 1: In assessing the effect of “Least Concern increasing”, it is unclear in the text whether includes or accounts for domestic species or not (lines 195-197).

The category of species referenced here refers to wild animals only and the counts in Table 1 exclude domestic species. We added wildlife or wild mammal throughout the text to clarify.

Table 1: Suggest present this from top to bottom in order of importance -i.e. as outlined in lines 187-189. Also, the final model structure is not given, and it is unclear whether the effects presented take into account the hierarchical structure of the data e.g. only threatened species are classified under “IUCN Criteria for Threatened Status”

The table was revised as suggested to include main effects in their order of relative importance in model fit as determined by the relative change in AIC for nested models with and without that factor.

The hierarchical structure of the model was clarified in Table 2. The IUCN criteria for threatened species categories were applied to threatened species only, and species that were not threatened were included in the model as not having any of these IUCN criteria apply. Categories of variables were modeled as dummy variables, coded as 0 or 1. Species not listed as threatened were assigned a 0 for criteria for threatened status categories, and species status was included as a main effect in the model. A more detailed description of the model and text specifying that only threatened species can have an IUCN criteria assigned were added to the Table 2 footnotes.

Line 303 - 304 - The wording here is helpful to clarify that effects being reported take into account other effects, but this is not mentioned elsewhere.

This text was added to the first mention of the final model results and a more simplified version of this wording was included in reporting of all other main findings.

Figure 2: It took me a while to see that the lines were connecting the animal silhouettes to the relevant circles for that species. A minor edit that might better highlight this would be to delete the grey lines after they pass (i.e. above) the silhouettes so it looks more like a line connecting to the silhouette.

It is also confusing to have these silhouettes overlapping with the (long) Population size reduction text. I suggest abbreviating this text to a few words in the figure, and have the full description in the Figure caption. I think it would also assist digestibility of the figure if the four categories in blue were also labelled A-D.

This figure underwent major revision as suggested by this reviewer and the handling editor. Revisions to Fig 2 include elimination of grey lines, matching line color to species color to better link the animal silhouettes to the relevant circles, and placing species icons on the same row. The text in the figure was reduced substantially and we make reference to the Guidelines for Using the IUCN Red List Categories and Criteria.

Lines 276 - 321 - Much of the text in this section is discussion, not results. It should be moved to the discussion.

We incorporated interpretation of some of the findings in the results section to improve clarity and remain within the journal's formatting and length requirements.

Discussion - By this point, taking into account the statement on lines 187 - 189 and the content in table 1, the take-home message I have inferred is that the research effort is the main determinant for detection of zoonotic viruses, followed by whether the species is domestic or not. Domesticated species contribute far more zoonoses than terrestrial mammalian wildlife species (lines 262-267). This should be the main finding, and any mention of the smaller effects of population size or IUCN status (which the authors are putting forward as the main finding) should be placed within that context. Of the fraction of zoonotic viruses that come from wildlife, significant effects of population size and threat status are evident, but how many viruses/how much risk does this represent in real terms?

We have revised the text by noting that we account for research effort when presenting each finding in the results section. Text was also modified to show that, overall, the majority of viruses were detected in wild animals, with rodents, bats, and primates implicated as hosts for 75.8% of zoonotic viruses described to date. The change in AIC was used to compare model fit for nested models with and without the variable of interest. In AIC value comparisons, main effects with a higher number of categories, such as population status categories (6 categories) and IUCN criteria (5 categories), will be penalized for complexity more than variables with only one category (domestication status) or continuous variables (In PubMed hits). Domestication status, IUCN criteria, and population status all had similar delta AIC values. Based on improved model fit for each parameter, all of the variables in the final model were needed to explain variation in number of zoonotic viruses and were significantly related to this outcome. We revised text to clarify presentation of the results in the context of multi-variable findings and the need to account for research effort, as detailed above.

Lines 334-340 - Species distribution, human distribution and habitats are not taken into account. The authors need to demonstrate the link between large population sizes and human contact, or explicitly state that this is a foundational assumption of these analyses. This is partly addressed Lin lines 341 - 342, but range overlap does not equate with habitat overlap

We agree; this sentence was revised to delineate our line of thinking and acknowledge this issue as follows; "Population range size similarly reflects opportunities for animal contact, and species

with larger ranges should have increased potential to overlap in range, and possibly share habitat with other species, enabling cross-species transmission and increasing risk of spillover to humans”.

Line 405-406: I do not agree that there is evidence to support that “abundant mammal species harbor more zoonotic viruses than less abundant species”. There has been greater opportunity for those to spillover into people, however the results for the threat categories show that that certain circumstances just allow us to be aware of the zoonotic viruses present in less abundant species.

Agreed, this was revised as follows; “We find evidence to support the premise that abundant mammal species have shared more viruses with humans . . .”

Line 409 - 410: Given the majority of zoonotic viruses come from domestic animals to humans, how much effect will this provide versus simply focussing on domestic animal health? There is no mention of the force of infection from domestic animals here.

This sentence was revised to highlight importance of domesticated species in spillover as follows; “Nonetheless, interventions aimed at ensuring biosafety in livestock production, minimizing interactions between wildlife and domesticated animals, and limiting contact with wildlife adapted to human-dominated landscapes . . .”

Additionally, I accept that it is difficult to study, and outside of the scope of this manuscript, but the number of zoonotic viruses from any given species does not reflect the burden of disease in people. The need for this information should be highlighted in the discussion.

We acknowledge the importance of this point and have added this idea to the discussion as follows; “Characterizing the burden of disease in people will assist in prioritizing in-depth, focused field studies that investigate the biological, ecological, and behavioral factors increasing risk of virus spillover for viruses with high impact on public health.”

Supplementary methods

Lines 8-10 - For completeness, add a date as to when the search was conducted, and the range of years of studies returned.

This search was initiated in December 2013 and all studies up through 2013 were included. This detail was added to main text methods, described in more detail in the supplemental information, and included in the text description of SI data file S1 and SI data file S2. All studies with data are also reported in the SI references within the data file S2.

Also, in the search terms, were wildcards included? e.g. zoonos* (to catch zoonosis and zoonoses). Was this list cross checked against existing databases (e.g. those on bat and rodent viruses) to see whether any were missed with this approach?

This list of viruses was cross-checked and we note references for cross-checking in the supplemental methods. Wild card search terms were not included, but we found cross checking with other lists of zoonotic viruses to be helpful in finalizing the list of viruses that met inclusion criteria for this study.

Lines S28-30 - Although it is stated in the supplementary materials, clarify here that evidence included PCR, virus isolation and serological evidence

Detection methods to classify a species as host are now stated and expanded upon in the SI. This detail was also added to the revised methods in the main text as follows "Among 142 zoonotic viruses that were examined, viruses with at least one mammalian host reported at the species level (n = 139) based on PCR, virus isolation, or serology were included in analysis (Data File S1.) Additional details regarding our literature search protocols and data inclusion criteria are in the electronic Supplementary Information (SI)".

Lines S69 - It is helpful if the Supplementary methods are readable on their own - Explain what the 'ZIP model' is.

The model was explained in the SI and ZIP abbreviation was removed.

Appendix B

UNIVERSITY OF CALIFORNIA, DAVIS

BERKELEY • DAVIS • IRVINE • LOS ANGELES • MERCED • RIVERSIDE • SAN DIEGO • SAN FRANCISCO

SANTA BARBARA • SANTA CRUZ

SCHOOL OF VETERINARY MEDICINE
WILDLIFE HEALTH CENTER
UNIVERSITY OF CALIFORNIA
(530) 752-4167
FAX (530) 752-3318
<http://www.vetmed.ucdavis.edu/whc>

ONE SHIELDS AVENUE
DAVIS, CALIFORNIA 95616-8734

February 5, 2020

Dear Professor Heersterbeek,

We have revised our manuscript RSPB-2019- 2736 "Global shifts in mammalian population trends reveal key predictors of virus spillover risk" based on suggestions from all reviewers. We thank the editor and the referees for their reviews and suggestions to revise the paper. We have carefully addressed each suggestion and made revisions to the manuscript text to reflect these changes.

Below, you will find a point by point response to all reviewers' comments (in blue italics). Please note that most suggested revisions required addition of text to the main paper. After making all suggested changes and resubmitting our manuscript, we were notified that we were over the page limit. We have thus revised down the content to be within the page limit. Changes suggested by the reviewers were retained, but some of the more detailed specifications related to our stepwise model building were moved to Supplemental Materials. As suggested by the journal, we have ensured that the main text can stand on its own with enough detail for a broad readership.

Thank you for the opportunity to improve our manuscript for publication by *Proceedings B*. Please contact me by email if you have any questions or concerns.

Sincerely,

Christine K. Johnson
University of California, Davis
ckjohnson@ucdavis.edu

Referee 1 Comments

Referee: 1

Comments to the Author(s).

Thank you for addressing all my comments.

Referee 4 Comments

Comments to the Author(s).

I have now read the manuscript entitled Global shifts in mammalian declines reveal key predictors of virus spillover risk by Johnson and collaborators, and submitted to Proceedings B, London.

I really enjoyed reading this very nice work, which is well written and reached sound conclusions based on clear and simple results. Reading the manuscript is easy, and the work in general is based on appropriate statistical methods, i.e., general linearized mixed modeling.

However, I have several concerns about the use of statistical methods and the context of their employment by the authors.

1. It is still hard to follow the statistical explanation even if the authors made an effort in this new amended version of their manuscript. I do understand that they used a forward regression procedure to reach a final model, i.e., a minimal model, but they poorly explained this in the document. I understand that the final model is minimal, best explained equation-s, but this should be clearly discussed. The authors should better explained what they really did for the reader (see following comment).

The statistical methods section was overhauled for clarity, including more overt descriptions of our dependent and independent variables in each step of the methods, as well our procedure to identify the minimal best explained equation as suggested.

2. Associated with this first comment, forward regression modeling is highly sensitive to the introduction of the first independent variable in the model, and in particular in the case where this-ese variable-s is-are non-continuous variable-s, e.g., ordinal, categorical, dichotomous. In general it is better to use forward and backward procedures separately, and then compare and discuss results obtained, or to use stepwise regression procedures. In addition, I am not sure that permutations of independent variables have been done in this work, since forward modeling is very sensitive to the introduction of the first variable. I would recommend the authors to do this proceeding to changing ranking order of independent variables introduced in the GLMM models, and see what happens.

We appreciate this comment - we re-evaluated the model building procedure to again evaluate alternative models and assess the best fit model as suggested. The final model was not sensitive to variation in variable selection and we arrived at the same final model each time. We thus clarified the approach described in the statistical analyses to ensure repeatability in our methods. In the main text Methods section, we added "Model building was initiated with log number of PubMed publications, and then variables were entered into the model using forward stepwise entry with all categories of a variable being entered at one time, starting with species status categories, then criteria for listing, then domestication status. Incidence rate ratios (IRRs) and their 95% confidence intervals (CIs) are shown for the final ZIP model (Table 2). Stepwise model building procedures are described in more detail in the Supplemental Materials. Parameter importance in improving model fit was assessed by removal of parameter groups one at a time, using ΔAIC ($AIC_{full} - AIC_{fitted}$) to compare to the best-fit full model (Table S1)." We also added an expanded section in the Supplemental Materials and refer to this in the main text "Stepwise model building procedures are described in more detail in the Supplemental Materials". In the Supplemental Materials, we added a section with header "Multivariable model selection" and text Multivariable zero-inflated Poisson (ZIP) regression modeling was used to factors related to zoonotic virus richness (sum of zoonotic viruses) in each mammalian species. Both terms for

reporting bias, log of the number of PubMed publications and data deficiency/unknown population trend, were evaluated in the ZIP model as an inflate variable (to predict excess zeros) and as a main effect variable, and selected based on optimization of model fit. Taxonomic order ($n = 28$ orders) was evaluated as a clustered random effect with robust standard errors and as a main effect to account for phylogenetic correlation among species within an order. The base model included log number of PubMed publications, and variables were entered into the model by forward stepwise entry with all categories of a variable being entered at one time, starting with species status categories, then criteria for listing, then domestication status. Backward stepwise elimination was used to remove categories of variables with $P < 0.3$. Two-way interactions between main effects that were significant in the multivariable model were similarly retained if significant ($P \leq 0.3$). Taxonomic orders were entered in a forward stepwise manner in the model without order as random effect. Variables were retained in the final model if statistically significant ($P \leq 0.05$), if they modified the coefficients of other covariates by more than 10% (indicating confounding), or if they improved overall model fit based on Δ deviance. Model sensitivity to order of variable inclusion through forward selection and backward elimination was evaluated by comparing terms retained in models using both approaches. We assessed change in AIC and BIC scores to compare nested models and arrive at a best-fit minimal model that included significant independent variables and best explained variation in the sum of zoonotic viruses in a species. Overall model fit for the final model and the alternate best model was reported as McFadden's R^2 . Competing multivariable models were evaluated by McFadden's R^2 and the Vuong test to compare overall model fit for non-nested models (Poisson, negative binomial, zero-inflated negative binomial, and other hurdle models). The best fit alternate model was a zero-inflated negative binomial model with log pubmed hits as the inflate variable, data deficient variable as a main effect. This alternate zero-inflated negative binomial model included all variables as significant effects, except the term for 'population size reduction by direct observation' was only marginally significant. The best fit alternate zero-inflated negative binomial model is shown in Table S2. We also show the best fit zero-inflated Poisson model without the variable representing number of publications in PubMed to indicate the influence reporting bias had on other variables related to virus richness in a species (Table S3)."

3. Final, minimal models retained are zero-inflated Poisson regression models, and the authors used other types of models e.g., zero-inflated negative binomial regression models as well, and I am curious to see the differences observed between the different models. This could appear as a supplementary material. Model fitting for this category of dependent variable distribution is generally difficult, with lack or difficulty of convergence, so presenting the different results obtained could be important for the reader.

We have included alternate models in the Supplementary Materials, including our best fit zero-inflated negative binomial regression model as well as the final zero-inflated Poisson model without the correction for reporting bias so that differences between models can be directly compared. We also added this sentence to the methods in the main text "We also show the alternate best fit model, a zero-inflated negative binomial model (Table S2), as well the final ZIP model without the term log number of PubMed publications (Table S3) to show model sensitivity to reporting bias."

4. The discussion/conclusion section should discuss about the spatial scale of this study and the results obtained, and replace them within the framework of hierarchical systems; determinants that are observed/obtained at one hierarchical level can be different for other scales. This is true for global species abundances as predictors of zoonotic virus species (so at global scale), but local mammal species abundance can be or not correlated to global mammal species abundance and appear not to be a good predictor of local zoonotic virus diversity. The authors should better discuss this point in this section.

We agreed spatial scale is a very important consideration. The discussion was revised to further highlight limitations in inferences with respect to local scale as follows "This study assessed inferences at a global scale

across all potential wildlife hosts for zoonotic viruses and determinants identified as predictors of zoonotic virus richness at this scale might not be related to zoonotic virus diversity in species at the local scale.”

As a reviewer, I would appreciate that the authors may reply to the above comments, notably point 2 which is the more objectionable.

Referee: 5

Comments to the Author(s).

General comments:

As other reviewers have highlighted, the article by Johnson and colleagues has involved a significant effort, from compiling a dataset to systematizing information about zoonotic viral infections across mammals. In my opinion, this article will be of broad interest for the Proc b readership and I anticipate that it could be highly cited because it establishes a link between species conservation and human health. I hope the authors find my comments useful.

The authors have done a great job responding to comments from previous reviewers. After carefully reviewing the previous version of the article and the author's responses my major concern with its current presentation comes from the lack of methodological details to allow for replication of the analyses and the presentation of some results, which require attention.

Since the major advance of this work is related to assessing the importance of conservation status and population trends of mammalian species in explaining the number of virus shared with humans without exploring mechanistic relationships between mammalian declines and virus spillover events, I find the title a bit confusing or misleading up to some point. I suggest rephrasing the title to reflect the findings of the article better.

Thank you for your review and comments; we addressed this suggestion by revising the title to “Global shifts in mammalian population trends reveal key predictors of virus spillover risk”. This precision in language relating population trends to conservation status was revised throughout the paper. We have found these comments very helpful in improving the paper.

Introduction

Lines 33 – 35. I find the phrase redundant in its terminology. Zoonotic diseases are by definition infectious diseases transmitted from animals to humans. I suggest rephrasing (e.g. “Infections that originate from animals and infect people (zoonotic diseases) comprise the majority...”).

This sentence was edited as follows . . .” Infectious diseases that originate from animals and infect people comprise the majority of recurrent and emerging infectious disease threats and are widely considered to be one of the greatest challenges facing public health”. First mention of zoonotic disease was moved to the second sentence.

Line 40 – 41. What do the authors mean by “disease detection capabilities...”? If it refers to the difficulties to detect infectious agents in wild species I suggest rephrasing, since the concept of disease refers to an altered physiological state experienced by an organism that affects its health not to the presence of an infectious agent in the organism.

We were referring to disease detection in a very general sense. This sentence was edited for precision as follows “. . . . hampered by pathogen detection limitations in wild species”.

Lines 41 – 42. I suggest providing support for why spillover events in humans are “massively under-reported”.

This description was rephrased to “Spillover transmission events in humans are likely vastly under-reported.” To provide support for this statement, another sentence was added “Spillover events could be easily missed, particularly in regions with limited access to healthcare, if the pathogen produces disease along a spectrum that includes mild and non-specific symptoms and there is limited or no human-to human transmission, thereby minimizing the number of people affected at any given time or place.”

Line 51. I find the concept of “range size” in the following sentence a bit confusing: “...detections of zoonotic diseases, such as larger range size and fecundity among rodents (7),...” Do the authors refer to home range sizes or distributional range? I suggest rephrasing to clarify the concept.

Range size was clarified throughout this paragraph to indicate these papers refer to ‘geographic range’

Line 77. I suggest changing the verb “predict” in the beginning of the line for “related” or alike to acknowledge that the model formulations used aimed to detect statistical relationships not actual predictive capacity of the explanatory variables considered.

Sentence was revised as suggested “. . . we show that species abundance and specific extinction threats are related to the number of viruses shared with humans across mammalian species”.

Materials and methods

Overall, I find the materials and methods difficult to follow. After reading the SI and the description of methods in the main text multiple times. A clearer description of the methods is crucial to allow for replication of the analysis and therefore results. It is also necessary since there is no code is associated to the manuscript for readers to replicate the analysis. In particular, the model formulations are confusing to me based on the narrative. For instance, from table S1, I understand that the whole dataset was used, however IUCN criteria for threatened species only applies to the subset of species (lines 52 – 67) with the levels of “VU, EN and CE” in the variable “Trends in Species Declines” (Table S1).

The methods section was revised to improve clarity. In particular we added a more detailed description of the handling of independent variables. The reviewer is correct that IUCN criteria only applied to species with threatened status. Dichotomous variables were created for each criteria and thus the criteria variables could be evaluated as modifiers of the effect of species status. All data, including the criteria categories for each species as analyzed in the study were included in data file S2.

I have a major concern about how the categorization of IUCN red list status and population trend (Table 1, and named as Trends in Species declines in table S1) were incorporated in the analysis. If I understand well, these variables were included in the models as categorical predictors ignoring their ordinal nature (e.g. commonly species would progress gradually in their categorization from the lowest to highest conservation concern (or vice versa). A potential approach to consider this is to include the conservation categories as numeric variables (and different transformations to take into account potential non-linearity) from, for instance 1 for the category with the lowest risk of extinction to 12 to the highest extinction risk (Critically endangered). In this way, the authors would explicitly test for the importance of conservation status in driving the richness of zoonotic viruses hosted by a particular species. The suggested ‘treatment’ of this variable is

further supported by the author's statement that conservation categories are proxies of species abundances (lines 103 – 105) which is of continuous numeric nature.

We thank the reviewer for this comment which has resulted in several important changes in the revised manuscript. To explain, we handled the variable trends in species declines as a main effect with ordered independent categories so that we could evaluate the relationship of each category with host virus richness in an unbiased manner. We did not assume that categories of species status would have a linear effect on species propensity to host zoonotic viruses. In fact, we assumed that the different categories of species status would have very different associations with zoonotic virus richness in terms of direction and measures of effect.

We are concerned about converting the species status categories to a numerical value because species status categories have very specific requirements according to the IUCN Red list. Also non-linearity would not be apparent to us in handling of the data. Furthermore, 2,218 species could not be included within a scale of species status because they were "data deficient".

We addressed this comment in the following ways:

Firstly, we renamed this variable that combines IUCN red list and population trends categories as "conservation status" throughout the manuscript for consistency. Table 1 was revised to reflect this variable name as "conservation status" and this header now matches the variable name in the model shown in Table 2. The "conservation status" categories were re-ordered in Table 1 to reflect their ordinal nature. Additional changes to the Methods included adding "Mammalian species were thus assigned levels in a new variable, conservation status, reflective of their combined measures of threatened status, population trend, and data deficiency (Table 1), which were included as dichotomous variables in the multivariable analyses."

Secondly, as suggested, we re-evaluated "conservation status" as a numerical variable for all species not classified as data deficient. Because this could only be done for a subset of data, we added this analysis to the Supplementary Materials as follows . . . "Linear correlation between species conservation status and zoonotic virus richness: We evaluated the linear relationship between conservation status and zoonotic virus richness by converting ordered categories of conservation status into a numerical value and assessing the overall correlation between conservation status and zoonotic virus richness in a species using the non-parametric Spearman's rho statistic, for species not described as data deficient (n = 3117). The numerical scale of conservation status had a positive linear relationship with the number of zoonotic viruses reported in a species. Specifically, species with increasing population trends had more zoonotic viruses (without data deficient species; rho = 0.215, two-sided P < 0.001)."

We also included a sentence in the Results in the main text to refer to this analysis "In an additional analysis of a subset of species that were not found to be data deficient, we found conservation status had a positive linear relationship with the number of zoonotic viruses reported in a species (Supplemental Materials)."

Since the analytical approach is complex in the sense that it uses multiple levels of categorization of variables (lines 51 – 67 in expanded methods in SI) I suggest adding a visual diagram (e.g. in the structure of a decision tree diagram) to explain the IUCN metrics and their adaptations included in the models. If the authors decide to include a diagram, I think it would be good to highlight the variables that were used for the different steps in the analysis (e.g. multivariable regression and ZIP).

To address this and a similar suggestion by the other reviewer, we expanded the description of the categorization of variables in the Methods section to specifically detail each step in the model building procedures. All variables described in the manuscript and presented in data file S2 were evaluated for inclusion in the multivariable ZIP model. The changes made to describe the analytical approach are describe above for Reviewer #4.

Considering the model selection approach implemented by the authors, the inclusion of the tables with the models at each step of inclusion of variables should be presented as supplementary information for better informing the readers with the behaviour of the different explanatory variables.

Description of model selection was modified based on suggestions from the other reviewer. We now also present an alternate best fit zero inflated negative binomial model in the SI, which illustrates the behavior of the explanatory variables under different modeling conditions. We also revised the Table S1 to show the changes to model fit with inclusion of each main effect in the final model described in the main text.

Lines 152 – 154: What do the authors mean by “Taxonomic order was evaluated as a clustered random effect ...” ? Were the ZIP models formulated including order as a random effect? If so, please clarify whether Order was included as random effect in all model formulations and phrase the model description accordingly (For instance: Line 148, A mixed effect zero-inflated regression modelling approach was used...”

Line 79 in SI: Is Taxonomic Order treated as a random effect? Consider previous comment about lines 152 – 154.

Taxonomic order was evaluated as a main effect and a random effect. This sentence was revised as follows “Taxonomic order (n = 28 orders) was evaluated as a clustered random effect with robust standard errors and as a main effect to account for phylogenetic correlation among species within an order.” We settled on including taxonomic order as a main effect once it was determined that several orders were significantly related to sum of zoonotic viruses. We thus further clarified that the mixed effect for order was dropped once order was considered as a main effect “ Taxonomic orders were entered in a forward stepwise manner in the model without taxonomic order as a random effect.” None of the models shown are mixed effect models.

Table S1 Overall the table organization is a bit confusing. I suggest to present the ΔAIC calculated as ($AIC_{fitted} - AIC_{full}$) in this way the sign of the ΔAIC would show direct relationship with “better or worse fit” as negative values would relate to lower AIC than the full model and therefore “better fit”. In addition, I suggest reorganizing the rows from the most supported model (full in this case) to the least supported one.

We revised Table S1 as suggested. Table S1 now presents ΔAIC calculated as ($AIC_{fitted} - AIC_{full}$) as suggested by the reviewer. We also have rearranged table rows in order to represent the most important variable (variable with largest ΔAIC) at the top of the table with decreasing variable importance. The table represents variable importance estimated as change in AIC after removal of the variable from the full model.

Results

Figure 1B: the inclusion of humans in the plot makes it difficult to visualize the other groups. I suggest removing the human species from the plot. This would help communicate how the abundance of other species relates to the proportion of total zoonotic viruses they host.

Given the relevance of humans to zoonotic disease transmission, we would like to include humans in Figure 1B to indicate human global abundance relative to the other species shown. However, analysis of the relationship between species abundance and proportion of zoonotic viruses was evaluated statistically by excluding humans to objectively measure the correlation between domesticated species abundance and number of zoonotic viruses. This analysis, as well as the analysis that includes humans, is described in the methods and results sections.

Figure 2: I really liked this figure. I suggest adding the scale (circles) at the bottom of the figure to facilitate interpretation by the reader without necessity of reading the caption while interpreting the figure.

We added a scale to the figure 2, which might not be much of an improvement to the figure. We have included the revised figure with a scale in the manuscript but think the figure is best presented as originally submitted.

Figure 3: I find this figure difficult to follow. In its current form, it does not allow the reader to visualize the level of connection (virus shared) among groups. I suggest removing the viruses as nodes in the network and instead use the thickness of the links to represent the number of viruses connecting the host groups (homologous to number of contacts among individuals in a social network), including humans. This would help readers visualize how well connected the different host species are but also their contribution to human infections individually and as groups.

We evaluated the one mode network with hosts as nodes and determined that far less can be discerned because of the high level of connectivity among hosts. The one-mode network is illegible due to the exponential increase in the number of edges between hosts. In the one-mode network, there is an edge between every host sharing at least one virus. For example, if one virus was reported in 10 hosts, in a two-mode network this is only 10 edges, but in a one-mode network this is 45 edges, i.e. a link between each of the 10 hosts. Because all viruses were zoonotic by definition, all hosts are connected to humans in the one-mode network and little else can be discerned. We thus resubmit our revised manuscript with the two-mode network to best illustrate the level of connectivity for viruses. We use centrality in the two-mode network to show a high level of connectivity between hosts (particularly as highlighted for domesticated species). We revised the figure to include this statement “Humans, who are host to all viruses, are not shown.”

Lines 184 – 185. I find confusing the use of percentages, are they the percentage of total zoonotic viruses hosted by those groups? Check the percentage for Chiroptera “(n=30%)”.

We agree; this sentence was revised as follows “In line with recent studies (6, 13), we found that the highest proportion of zoonotic viruses were reported among species in the orders Rodentia (61%), Chiroptera (30%), Primates (23%), Artiodactyla (21%) and Carnivora (18%), and fewer viruses were detected in other mammalian orders (Fig. 1).”

Lines 199 – 202. This sentence (if required) should be included in section “Materials and Methods”.

These sentences were deleted due to redundancy with text in the methods.

Lines 230 – 239. A visualization of the ‘dose response’ relationship reported would help better report the results.

We now include a direct assessment of the linear relationship between conservation status on a numerical scale and the number of zoonotic viruses as suggested in the Supplementary Materials, which further illustrates the dose response relationship. We also revised this section of the results to better explain the ‘dose response’ relationship as follows . . .” After adjusting for all other factors, we detected a dose-response type relationship between increasingly threatened conservation status and a corresponding decrease in the number of viruses animal species share with humans, as evidenced by the gradual decrease in incidence rate ratios as species conservation status changes from least concern increasing to critically endangered (Table 2).”

Lines 248 – 257: This section presents findings from other studies and hypothesis from the authors. I suggest moving them to the discussion section.

Lines 263 – 271. As in the previous comment, this section should be moved to the discussion since the authors discuss results and present their hypothesis about transmission of pathogens between humans and highly threatened species.

Lines 279 – 281. As previous comment for lines 248 – 257.

Lines 286 – 297. As the previous comment about lines 248 - 257.

Lines 346 – 361. As the previous comment, I find these lines to be material for the discussion.

We combined the results and discussion in this manuscript to improve readability as we have seen in published Proceedings B articles. To indicate that this is the style we are using, we renamed this section “Results and discussion”.

Line 273. I find this subtitle confusing. The subtitle refers to a “positive feedback” between the driver of mammalian declines and the number of viruses shared with humans, however this specific loop effect was not assessed. It is also not clear if the ‘effect’ reported in this section and table 2 (under the “IUCN Criteria for Threatened Status subsection) was tested in a sub-model that included only threatened species (?), if so, please rephrase sections accordingly.

This subtitle was changed to “Convergence in drivers for mammalian species declines and zoonotic virus richness”

Line 278. What do the authors mean by “given all other factors in the multivariate model”? Please rephrase the sentence for clarification.

This sentence was rephrased as follows “After adjusting for other significant effects in the multivariable model, we find that threatened species with a population size reduction due to exploitation (IUCN Red List category A1-4(d), n = 256 species) have over twice as many zoonotic viruses as compared to threatened species listed for other reasons (Table 2).”

Lines 283 – 286. I find the reference to figure 2 confusing. The sentence refers to the model outcomes (“predictions”) which are the coefficients, therefore I was expecting Figure 2 to show the model estimates (coefficients). However, the figure shows the ‘raw’ number of viruses shared between different taxa (by conservation status) and humans. I suggest referring to figure 2 to report the raw number of zoonotic viruses by group and maybe, include a replicate of figure 2 reporting the model estimates.

We made an error in referring to figure 2 here, and rather meant to refer to Table 2. This mistake was corrected.

Discussion

Overall, I think a more direct link between the subheadings in the results section, and the narrative in the discussion is needed. For instance, the discussion does not mention the role of primates and bats in the number of viruses shared with humans, which was presented in the results as a subheading. Please ignore this comment if lines 346 – 361 are included as discussion material.

We revised the headings to include discussion with results. This section is now labelled “conclusions and future directions”.

Line 381: I find the term “community size” a bit confusing. Does this refer to population size (same species of hosts) or the actual assembly of host species? I think that the first would be more appropriate for the context (?).

This sentence was revised to facilitate readability to a broader audience as suggested “Reservoir populations have a critical population or community size required for infectious disease transmission”.

Lines 397 – 403. Please add references to these statements.

This finding is based on the data presented in the paper and our inferences, so this was rephrased as follows “Many of species listed as least concern with increasing abundance trends by the IUCN Red List are adaptable wild mammalian species that have benefitted from a close relationship with humans. These species could have habitat and dietary niches that overlap with humans in dwellings or in agricultural practices, further enabling direct and indirect contact with similarly adapted sympatric species, domesticated species, and humans.”

Line 422: The assumption that a species is a competent host for spillover because it was seropositive is misplaced. An individual/species could be seropositive (exposed) but not a competent host (i.e. suitable for transmission), or not be of epidemiological relevance as a ‘source’ of infection. The authors must acknowledge how this assumption in the treatment of serology and PCR data could influence their inferences. One potential consequence could be an overestimation of species as a “source” of pathogens, especially in cases where multiple species may seroconvert, but only a few are main shedders. For example, in the case of Hendra virus in Australia, although *Pteropus poliocephalus* and other pteropid bats may be seropositive against the virus (supporting exposure), PCR results and viral isolation support that *P. alecto* and *P. conspicillatus* are the main reservoirs (see Field HE (2016) Hendra virus ecology and transmission, Current opinion in Virology, 16:120-125, and references). Based on this case (Hendra virus), the list of hosts presented in the supplementary information seems far-reaching and unsupported.

We concur with this comment and have carefully stated our assumptions and the reasoning for these assumptions. We further revised our description of the data in the SI as follows “The dataset of animal species associated with zoonotic viruses in this study is provided in Data file S1, which should be considered baseline data pending further investigations to confirm wild animal species as competent reservoirs or determine that some suspected host species are not epidemiologically important sources of spillover to humans. We propose re-evaluating epidemiological models of spillover risk in light of new data.” In the main text, but especially in an expanded section in the SI, we compare inferences that can be made from molecular vs serological evidence. We also note that we have modeled the inclusion and exclusion of host serology data for flaviviruses and determined that this did not affect classification of which wildlife species were likely to host these viruses. Similarly, Olival et al. as referenced found that including or excluding serological data when quantifying virus richness per mammalian species did not alter predictions of how host and virus traits affect patterns of virus spillover from wild mammals to humans. Both over- and under-reporting of hosts for viruses remain a concern but we have drawn inferences following objective treatment of the data available and have been transparent about the nature of the data.

Line 431 - 432. I suggest adding a table similar to table S1, including the model formulations with and without the number of publications as variable. In this way, the authors would support the statement about the explanatory importance of the different variables being consistent.

As suggested, we included the model without number of publications (InPubMed hits) in the Supplemental Materials. This model is now fully described in Table S3 to enable direct comparison of explanatory variables with and without this variable used to adjust for reporting bias.

Line 432. This line refers to the statistical behaviour of variables in the model formulations, however the reference included (18) is inappropriate. The reference is to the Red List of the IUCN.

Corrected; we now refer to Table S3.

Lines 437 – 439. Could the authors make some suggestions or recommendations for how “investigator-driven” studies could build better data for understanding global patterns? It would also be useful to add an opinion or view about what aspects, processes or inferences could be gained from continuous updates or re-evaluation of models.

We substantially expanded this last paragraph in the conclusion as recommended, particularly in light of this and previous suggestions made by this reviewer.

References

References 8, 9, 41 are not referred to in the text.

Corrected, thank you